# A Minimal Agent for Automated Theorem Proving

**Borja Requena** [1]   **Austin Letson** [1]   **Krystian Nowakowski** [1]   **Izan Beltran-Ferreiro** [1]   **Leopoldo Sarra** [1]

## Abstract

We propose a minimal agentic baseline that enables systematic comparison across different AI-based theorem prover architectures. This design implements the core features shared among state-of-the-art systems: iterative proof refinement, library search and context management. We evaluate this agentic approach using qualitatively different benchmarks and compare various frontier language models and design choices. Our results show competitive performance compared to state-of-the-art approaches, while using a significantly simpler architecture and a fraction of their cost. Additionally, we demonstrate consistent advantages of an iterative approach over multiple single-shot generations, especially in terms of sample efficiency and cost effectiveness. The implementation is released open-source as a candidate reference for future research and as an accessible prover for the community.

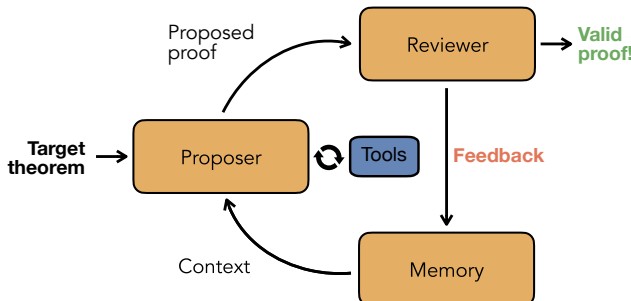

*Figure 1.* **Minimal agent for theorem proving.** The design focuses on three main aspects: iterative proof refinement, memory system, and access to tools. A proposer agent writes Lean code to prove a given theorem. Then, a compiler verifies if the proposed proof works. If so, it is double-checked by a reviewer agent to prevent any form of cheating. If the code does not compile, or the reviewer objects, the feedback is sent to the memory module, and the cycle starts over with the proposer refining its previous proof informed by the additional context provided by the memory module. In addition, the proposer can be given access to tools such as library search or web search, and call them a fixed number of times before giving its proposal.

## 1. Introduction

Automated theorem proving represents an avenue for verifiable scientific reasoning in artificial intelligence (AI). It can enable the validation of proposed arguments and their conclusions through a rigorous formalization, typically in the form of a theorem that, if proven, certifies the correctness of the arguments without the need of human intervention. This is a great opportunity for the research community in general: in the long run, not only can it speed up the verification of new theoretical results produced by mathematicians and scientists, but it can also help grounding generative models into logically consistent arguments, possibly enabling automated scientific contributions.

A very popular interactive theorem prover, especially in

the AI and math community, is Lean (Moura & Ullrich, 2021). This programming language has been used in several mathematical formalization projects (Becker et al., 2025; Commelin & Topaz, 2023), as well as in AI research projects that demonstrate advanced mathematical reasoning (Hubert et al., 2025; Achim et al., 2025). However, formal methods are underused despite being powerful tools with broad applicability in scientific and engineering domains, such as theoretical physics (Tooby-Smith, 2025; Székely, 2010), control theory (Jasim & Veres, 2017; Rashid & Hasan, 2017), or even quantum algorithms (Peng et al., 2023; Lewis et al., 2023). Recent advances in AI tools for argument formalization (Lu et al., 2024; Poiroux et al., 2025b) and theorem proving (Varambally et al., 2026; Ren et al., 2025; Chen et al., 2025b; Achim et al., 2025) lower the barrier to entry and make theorem provers more accessible to a broader community. Nevertheless, these AI systems are often quite complex and difficult to adopt. When open sourced, they typically require setting up a large-scale infrastructure to deploy them, and when based on off-the-shelf proprietary models, they tend to be expensive. For larger-scale practical adoption of AI-based formalizers and theorem provers, affordability and easiness to use are key.

---

[1]Axiomatic AI, Barcelona, Spain. Correspondence to: Borja Requena <borja@axiomatic-ai.com>, Leopoldo Sarra <leopoldo@axiomatic-ai.com>.

*Proceedings of the 43rd International Conference on Machine Learning*, Seoul, South Korea. PMLR 306, 2026. Copyright 2026 by the author(s).

Currently, several approaches for AI-based provers are being explored, with a particular emphasis on competition math as a proxy for evaluating mathematical reasoning capabilities (Tsoukalas et al., 2024; Zheng et al., 2021; Letson et al., 2026). Many works assemble together several complex components, including fine-tuning based on reinforcement learning on large-scale synthetic datasets (Lin et al., 2026; Wang et al., 2025a; Dong & Ma, 2025), enforcing supervised fine-tuning on proof traces (Xin et al., 2025; Hubert et al., 2025), or proposing agentic architectures with advanced recursive decomposition and several tools (Varambally et al., 2026; Chen et al., 2025a).

While these systems demonstrate great performance on the established benchmarks, they may quickly lose their practical applicability due to the fast release cadence of new versions of Lean and Mathlib (mathlib Community, 2020), a formal mathematical library for Lean. Moreover, unlike a few years ago, when general-purpose large language models (LLMs) had limited ability to write Lean code and Lean-specific fine-tuning was often necessary, recent state-of-the-art models have made significant progress in closing this gap. The constant improvement of LLMs makes it hard to disentangle the source of performance improvements between different provers: do we observe a better performance thanks to an architectural innovation, or is it mainly due to using a more capable LLM? This is a critical distinction to make when trying to improve on the state of the art.

In this paper, we propose AxProverBase, a minimal agent for automated theorem proving. Its design is shown in Figure 1. We identify iterative proof refinement, a context management mechanism, and the ability to search as the current main sources of success in the field, in addition to the choice of the base model that proposes a proof. Thus, we design a modular architecture that features these main components, which is simple enough to allow us to conduct multiple ablation studies as we build it from the bottom up to clearly understand their impact on the overall performance. We show that:

- The ability to perform iterative proof refinement is the biggest factor for the performance and it alone is sufficient to outperform many complex state-of-the-art approaches.

- Memory mechanisms significantly reduce the number of errors, yielding the second largest performance boost.

- Tools to search through Lean libraries like Mathlib are helpful, although not nearly as impactful as the previous two points.

- More powerful models see the largest performance improvements when provided with the right scaffolding. Capitalizing on their capabilities pays off.

- Simple agentic approaches, like AxProverBase, can already achieve competitive results across domains, making them an appealing accessible option for regular use in real projects.

We open source this agent together with the evaluation infrastructure to encourage its use as a baseline model that naturally improves over time with the progress of LLMs, and that can be improved upon by enhancing each of its main parts. In addition, the proven effectiveness and simplicity of this approach may also encourage its adoption by the theorem-proving community.

**Conflict of interest disclosure** All authors are employed by Axiomatic AI, which developed AxProverBase, the system introduced and evaluated in this paper. The implementation is released open source. The authors are not aware of any other financial conflicts of interest related to this work.

## 2. Related work

### 2.1. Automated Theorem Provers

In recent years, two main directions have emerged in automated theorem proving: tree-search methods (Xin et al., 2025; Wang et al., 2025b; Yang et al., 2023; Li et al., 2025; Shen et al., 2025; Wang et al., 2023; Lample et al., 2022; Achim et al., 2025; Hubert et al., 2025) and whole proof generation (Lin et al., 2025; 2026; Chen et al., 2025b;a; Varambally et al., 2026; Breen et al., 2025; Baba et al., 2025; Dong & Ma, 2025; Cao et al., 2025; Ren et al., 2025). Tree-search methods construct the proof line by line, interacting with the Lean environment to inform subsequent proof steps. In contrast, whole proof generation approaches attempt to close the proof providing all the code at once before compiling.

**Tree-search methods.** Tree-search methods, have recently achieved significant milestones in automated theorem proving. AlphaProof (Hubert et al., 2025) is perhaps the most prominent example. It achieved a silver medal at the International Mathematical Olympiad 2024, scored 56% on PutnamBench, and saturated the MiniF2F benchmark. REAL-Prover (Shen et al., 2025) integrates a system to retrieve semantically relevant theorems from Mathlib at each proof step. This addition showed a clear advantage over previous tree-search methods, improving performance on the FATE-M benchmark by over 10%. Aristotle (Achim et al., 2025) introduced interleaved informal reasoning moments and a specialized geometry-solving engine to a tree-search-like method.

**Whole-proof methods.** Whole-proof methods, on the other hand, are currently dominating the PutnamBench

leaderboard (Tsoukalas et al., 2024; 2026) (see Appendix A). There, the leading provers are agents that generate full proofs combined with an iterative proof refinement strategy based on the Lean compiler feedback, and have access to various Lean tools, especially to search Mathlib.

A central theme has been how to obtain useful training signal despite the sparsity of correct formal proofs. Self-play theorem provers (Dong & Ma, 2025) addressed this by coupling a prover with a conjecturer that generates synthetic, barely-provable problems, creating an adaptive curriculum. DeepSeek-Prover-V2 (Ren et al., 2025) advances this direction at scale by decomposing complex theorems into subgoals and convert successful subproofs into chain-of-thought supervision for reinforcement learning. Goedel-Prover-V2 (Lin et al., 2026) then incorporated feedback from the Lean compiler for proof-refinement, setting the state of the art for that time on PutnamBench and MiniF2F (Zheng et al., 2021) with models that were 20 times smaller than DeepSeek-Prover-V2.

Among open-source models, Hilbert prover (Varambally et al., 2026) is currently leading the PutnamBench Leaderboard (Tsoukalas et al., 2026), with roughly 45.9% pass@1 and 70% pass@1840 success rate. Hilbert combines informal reasoning, theorem retrieval, specialized Lean proof generation, and verifier feedback in a recursive proof-construction loop. When direct proof generation fails, it produces a verified proof sketch and extracts its intermediate subgoals to be proven independently. Unresolved subgoals are recursively decomposed until a complete proof can be assembled.

Seed Prover V1.5 (Chen et al., 2025a) integrates most of the aforementioned techniques into one system: iterative refinement, structured compiler feedback, library search, lemma decomposition, draft-sketch-prove approach (Jiang et al., 2023), complex context management, and even a geometry-solving engine. It achieves excellent performance on PutnamBench and FATE benchmarks.

Ax-Prover (Breen et al., 2025) is based on a general-purpose model with tools to interact with files, inspect and diagnose Lean code, and search relevant libraries in a ReAct (Yao et al., 2023) architecture where the agent can decide when to receive compiler feedback. This agent outperforms special-purpose provers and generalizes to multiple domains beyond competition math, showing the potential of using frontier LLMs in a simple iterative proof-development framework. We draw inspiration from these general ideas for this work.

Finally, the current leader on the PutnamBench leaderboard is Aleph Prover (Logical Intelligence, 2026), but there are no public papers or technical reports that describe its agentic architecture in detail.

## 2.2. Benchmarks

Currently, some of the most popular benchmarks for AI provers consist of problems from real-world math competitions formalized into Lean code by human experts. In particular, MiniF2F (Zheng et al., 2021) includes the Math-Olympiad competitions, and PutnamBench (Tsoukalas et al., 2024) collects problems from the Putnam undergraduate competition.

More recently, the focus is gradually shifting towards more advanced mathematical fields. For instance, FATE (Jiang et al., 2026; Shen et al., 2025) gathers abstract and commutative algebra cases that reflect the character of modern mathematics research. It is split into three distinct difficulty levels: M, H and X. While FATE-M has presented a modest challenge for popular open-source provers, which can solve around 50% of problems (Jiang et al., 2026), FATE-H and FATE-X have been well beyond their reach, with 3% and 0% of the problems solved, respectively. Another great example is LeanCat (Xu et al., 2025), which targets category-theoretic formalization: a unifying language for mathematical structure, and a core layer of modern proof engineering.

## 3. Architecture

We propose a simple and modular architecture for a theorem proving agent with three main core features: a feedback mechanism for iterative proof refinement, a memory system to preserve information between iterations, and access to tools to search the necessary information to complete the proof. The iterative refinement process is based on a loop in which an agent proposes a proof that can be checked and returned with concrete feedback about its shortcomings. Thus, we identify three main modules in our architecture: a proposer agent, a review system, and a memory system, depicted in Figure 1. This modularity allows us to independently modify the building blocks to conduct ablation studies.

**Proposer agent.** The proposer's goal is to complete a proof by writing Lean code. It receives the target theorem within its associated context, such as the full file content, and information about its previous attempts at proving the theorem from the memory module. With this information, the prover proposes a proof to complete the target theorem.

The proposer can take any form, from a specialized Lean model, to a general-purpose LLM, or even a deterministic tactic. In our implementation, we consider a ReAct-style agent (Yao et al., 2023), using a general-purpose LLM with the optional possibility to use tools. When tools are available, the proposer can make a single round of parallel tool calls before generating a proposed proof. We consider the

following tools:

- **Library search.** A custom deployment of LeanSearch (Gao et al., 2024), which uses vector embeddings for premise selection from Mathlib.

- **Web search.** Tavily (Tavily, 2025) allows the agent to find proof strategies. We emphasize that we enable web search because we focus on practical usefulness and addressing the challenge of writing Lean code that compiles, which is the main challenge and bottleneck in the field (Jiang et al., 2026), rather than reasoning through the solution informally.

**Review system.** The review system has two components: a compiler and a reviewer agent. The compiler programmatically verifies that the proposed code proves the target theorem by compiling it. If the code does not compile, it returns a feedback message containing the compilation errors. Otherwise, it checks for `sorry`/`admit` placeholders in the code and returns a feedback message with goal states extracted at their locations using LeanInteract (Poiroux et al., 2025a). This allows the agent to develop its proof incrementally through subsequent iterations. If the code compiles and does not contain any `sorry`, additional axiom, or suggestion tactics (like `apply?`), the compiler sends the proposed code to the reviewer agent.

The reviewer's main role is to verify that the target theorem statement has been preserved and to flag potential cheating mechanisms not covered by our previous checks. These may include subtle cases such as metaprogramming tricks that can make the code compile without providing a complete proof of the intended theorem. Hence, the LLM-based reviewer acts as an additional safety layer on top of the deterministic checks for known loopholes.

**Memory.** The memory node provides the proposer with context from previously failed proof attempts. This mechanism can take multiple forms, ranging from simply keeping the full transcript of interactions, to a multi-stage retrieval system. An important factor to consider is keeping a manageable context size for the proposer. For instance, keeping the complete history can quickly bloat the context, leading to slower and more costly LLM calls that would gradually deteriorate in performance, and eventually exceed the model's context length. Here, we compare the following implementations:

- **No memory**: The proposer receives no context from past proof attempts.

- **History of previous $n$ attempts**: The proposer receives the $n$ most recent proof attempts, each containing the full reasoning, code and feedback.

- **Self-managed context**: We implement a self-reflection strategy (Shinn et al., 2023; Pan et al., 2023; Renze & Guven, 2024) in which the agent manages a short context for itself where it can write relevant information, similar to a lab notebook. After every iteration, we let the agent reflect on its proof attempt with access to its reasoning, code, feedback, and managed context. The agent updates the notes with new key technical insights, while preserving important past lessons to prevent repeating mistakes.

We provide all the prompts for each component of our architecture in Appendix B, and a full trace of an illustrative example in Appendix C.

## 4. Experiments

We consider the three main elements of our architecture: feedback, memory, and tools, and conduct experiments to understand their impact on the prover performance. Then, we study the effect of the underlying foundation model by embedding several LLMs with different capabilities into a fixed agentic framework. Finally, we analyze the cost of each proposal compared to their performance and select an architecture that we evaluate on standard benchmark datasets.

We conduct all the ablation studies on a subset of the PutnamBench dataset (Tsoukalas et al., 2024) composed of 100 randomly selected samples that we list in Appendix D. This is motivated by two main reasons: reducing the overall cost of this study, which in turn allows us to conduct more experiments, and preventing the overfitting to a particular target dataset, which can compromise the integrity of the final performance evaluation and the generalization to other data sets.

### 4.1. Bottom-up system analysis

We build our system from the bottom up to understand the significance of each element in the architecture. We consider the following configurations in order of complexity.

**Single shot.** Starting from a bare-bones LLM with no tools, feedback, or memory, we measure the pass@$k$ performance with an increasingly higher number of samples $k$. In this setting, pass@$k$ is the fraction of theorems for which at least one of $k$ generated proofs is correct. The success rate increases with $k$, although it is a rather inefficient strategy, as shown in Figures 2 and 3. With a very low single-shot success rate, illustrated by pass@1, additional independent proof samples barely yield any significant improvements. Indeed, we find that the most difficult theorems are hardly-ever proven with this strategy, and only in the easier cases we see the benefits of multiple sampling.

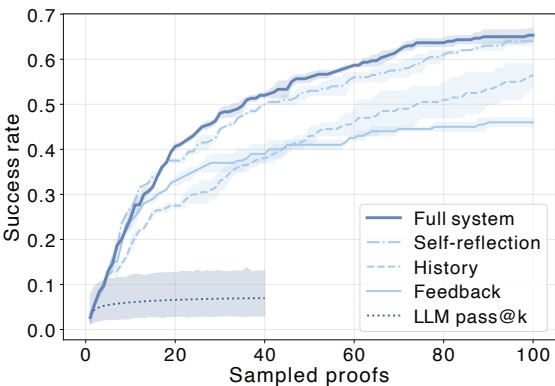

*Figure 2.* **Ablation study on different components of the minimal agent**. We compare single-shot parallel calls of a general-purpose LLM on its own and embedded within the proposed agentic framework. We show the performance as a function of the number of sampled proofs progressively incorporating elements of the architecture described in Section 3. Starting by adding a feedback mechanism, we then evaluate two different memory mechanisms (history of previous attempts and a context managed through self-reflection), and we finally incorporate search tools. The entire experiment is conducted with Claude Opus 4.5 with a thinking budget of 10k tokens. The pass@$k$ values are computed out of 50 independent realizations, and we show the 95% confidence interval around them. For the iterative approach, we compute the mean percentage of proven theorems at each iteration and show the standard error of the mean around it. We average over two independent realizations for the different ablations, and three for the full system.

**Iterative refinement.** To increase the sample efficiency, we introduce a feedback mechanism to inform the subsequent iteration after a failure with information about the errors. This is implemented with the simplest memory mechanism, comprised of the history of the previous $n = 1$ attempts, as described in Section 3. This allows the agent to iteratively improve on its initial proof proposal, yielding the largest performance gain out of any other element in our architecture, as it enables the reliable development of more complex proofs. Nevertheless, we observe diminishing returns in performance as we increase the number of iterations, as we see in Figure 2, due to the limitations of this simple approach, such as being prone to repeating past errors.

We distinguish between refinement iterations within a single prover run and the number of independent runs used for pass@$k$ evaluation. The iteration budget within a single prover run controls the amount of test-time compute spent developing one final proof outcome: either a completed proof or a failure after the budget is exhausted. This is qualitatively comparable to the prover attempts and recursion depth parameters of Hilbert (Varambally et al., 2026), or to the "light"/"medium"/"heavy" test-time scaling parameter of Seed-Prover (Chen et al., 2025b). By contrast, the $k$ in pass@$k$ counts independent prover runs on the same prob-

lem, each with potentially multiple iterations. Therefore, Figure 2 shows the pass@1 performance as a function of the number of iterations (except for the single-shot LLM curve).

**Memory.** We mitigate some of the system's limitations by introducing a longer-term memory mechanism, which allows the agent to keep track of relevant information from past proof attempts. We explore two approaches: a history of the previous 5 past attempts, and a self-managed context, described in Section 3. Both strategies yield significant performance improvements over the immediate iterative refinement, as we show in Figure 2. They allow the agent to spend more iterations addressing open goals to develop the proof by reducing the number of Lean errors it makes, as we detail in Appendix E. Letting the agent manage its own context yields better results in terms of performance, cost and stability, proving on average 7% more theorems at a 20% lower total cost (10% for equal number of iterations) with half the dispersion between runs. Thus, we settle for this memory management system.

**Tools.** We identify that the largest sources of errors in the proposed proofs are related to misidentifying or misusing Mathlib (mathlib Community, 2020) theorems and tactics (see Appendix E for details). We provide the agent with tools to search across the Mathlib library through our own LeanSearch implementation (Gao et al., 2024) (see Section 3), as well as to look for information through the internet with Tavily search (Tavily, 2025). We achieve the best performance with the addition of the search tools on top of the feedback and memory, although the improvement is not as significant as the previous two foundational elements of the agent (Figure 2).

### 4.2. Foundation model comparison

A core component of agentic provers is their underlying language model. While state-of-the-art LLMs are becoming increasingly better across a wide variety of tasks, we still find discrepancies in their capabilities on specialized tasks, such as theorem proving in Lean. We conduct multiple experiments comparing foundation models on their own and embedded in our agentic framework to get a better understanding of how they can benefit from such infrastructure and the impact they have on the final performance. In these experiments, we set the maximum number of iterations of our system to 50, use a self-managed memory, and search tools.

**LLM model comparison.** We compare the performance of several LLMs both single-shot and within the AxProver-Base framework. For single shot generation in Figure 3(a), we can observe a rather homogeneous performance between

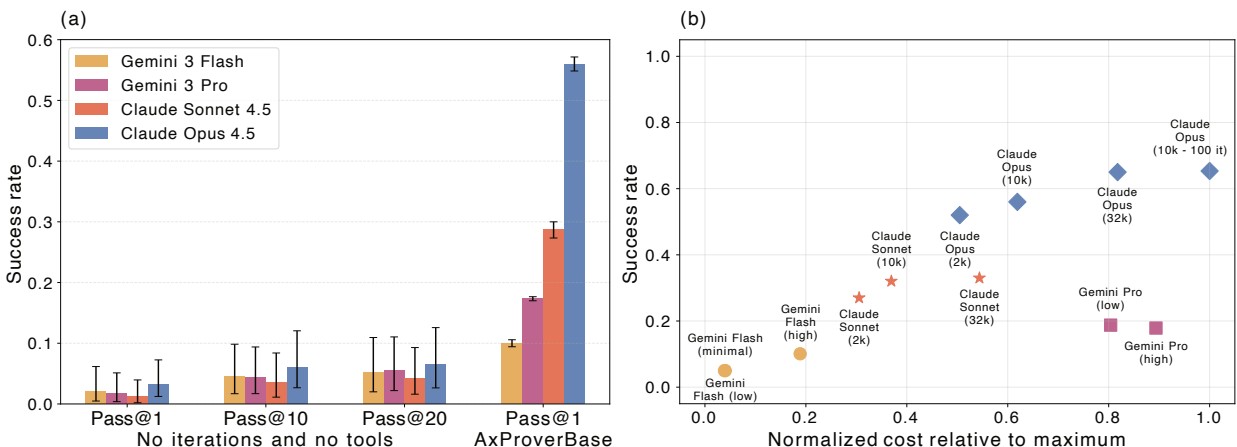

*Figure 3.* **(a) Comparison between LLMs**. Claude Sonnet 4.5 and Opus 4.5 have a thinking budget of $10,000$ tokens. Both Gemini 3 Flash and Pro have thinking set to "high". The pass@$k$ are computed from 50 independent proof samples for each model and we show the 95% confidence interval around the obtained value. For the full system, we show the mean performance over 3 full runs up to 50 iterations and show the standard error of the mean. **(b) Cost-performance analysis**. We compare the performance of different language models as we vary their thinking budget with respect to their cost. We consider thinking budgets of 2k, 10k and 32k tokens for both Claude models, and "low" and "high" for the Gemini models. We also test the "minimal" option for Gemini Flash.

the stand-alone models, with pass@20 completing around 5% of tasks for all of them. However, we observe stark differences when used with our agentic framework.

First of all, we observe a significant performance improvement across all models with respect to their stand-alone version, as we have already discussed in Section 4.1. Additionally, as shown in Figure 3(b), "smarter" models, such as Claude Opus or Gemini Pro, show a relatively greater improvement in performance than their "less smart" counterparts. Hence, a simple agentic system can act as an enabling framework for its underlying LLM to make the most out of its capabilities.

Interestingly, we observe significant differences between Claude and Gemini models, with Claude models proving about three times more theorems than their Gemini counterparts. A deeper analysis of their traces reveals a clear tendency of Gemini models to import and use elements from older Lean and Mathlib versions and even use incomplete or hallucinated imports, despite having the versions specified in the prompt and having access to search tools. We provide an in-depth error analysis in Appendix E where we show that 56% and 33% of the errors from Gemini 3 Flash and Pro are due to bad imports, in contrast with 1.1% and 0.4% for Claude Sonnet and Opus 4.5. These types of mistakes waste the iteration budget and remove resources from the actual proof development.

**Thinking budget.** Observing that stronger models gain greater advantages from this framework, we further investigate how the choice of the underlying model affects performance by exploring a wide range of model capabilities through various models and configuration settings. We do so

by adjusting the thinking budget parameter of these LLMs to increase or decrease the amount of tokens that are invested in their chain of thought before producing an answer. In Figure 3(b), we show the relationship between the cost and performance of each model as we vary their thinking budget. For the cost, we consider the average money spent to evaluate each of the prover instances in a full experiment. This serves as a reference quantity to compare models with different capacities that produce different quantities of input and output tokens. We normalize the costs with respect to the most expensive approach to perform a relative comparison between them. As expected, higher thinking budgets generally lead to higher costs and performance, although we observe that some models benefit more than others. Gemini 3 Flash, Claude Sonnet and Claude Opus 4.5 can benefit significantly from an increased thinking budget. In contrast, Gemini 3 Pro does not show significant differences in performance between "high" and "low" settings, just like Gemini 3 Flash between "minimal" and "low" thinking modes. We observe that Sonnet's performance saturates as we increase the thinking budget from 10k to 32k tokens at high thinking budgets, while Opus shows a steady positive trend. This budget increase in Opus allows it to achieve the same performance as doubling the number of iterations, at a significantly lower cost.

### 4.3. Benchmark dataset evaluations

We select the model performing best in the experiments in Sections 4.1 and 4.2: Claude Opus 4.5 with a thinking budget of 32,000 tokens and 50 iterations, and evaluate it on a series of popular benchmarks. We assess its competition math capabilities with the PutnamBench data set (Tsoukalas

| Model | PutnamBench | FATE-M | FATE-H | FATE-X | LeanCat |
|---|---|---|---|---|---|
| DeepSeek V2 | 7.1% @1024 | 25.3% @1
62.7% @64 | 0.4% @1
3.0% @64 | 0.0% @64 | 9.0% @32 |
| Goedel Prover V2 | 13.0% @184 | 21.2% @1
48.7% @64 | 0.4% @1
2.0% @64 | 0.0% @64 | 5.0% @32 |
| Kimina Prover | 1.5% @192 | 10.5% @1
36.0% @64 | 0.5% @1
2.0% @64 | 0.0% @64 | 2.0% @32 |
| Seed-Prover 1.5 | 87.9% | – | 80.0% | 33.0% | – |
| Claude Opus 4.5 | – | – | – | – | 8.25% @1
12.0% @4 |
| REAL-Prover | – | 56.7% @64 | – | – | – |
| Ax-Prover | 13.8% @1 | – | – | – | – |
| Hilbert | 55.9% @1
70.0% @1840 | – | – | – | – |
| **AxProverBase**
(Opus 4.5, 32k, 50 it) | 54.7% @1 | 98.0% @1 | 66.0% @1 | 24.0% @1 | 59.0% @1 |

*Table 1.* Comparison of various AI-based theorem provers with AxProverBase using Claude Opus 4.5 with 32k thinking token budget and 50 iterations. Results for other provers are as reported either in their respective papers or the dataset papers: DeepSeek V2 (Ren et al., 2025), Goedel Prover V2 (Lin et al., 2026), Kimina Prover (Wang et al., 2025a), Seed-Prover 1.5 (budget of 10 H20 days / problem) (Chen et al., 2025a), Claude Opus 4.5 (LeanCat evaluation (Xu et al., 2025)), REAL-Prover (Shen et al., 2025), Ax-Prover (Breen et al., 2025), and Hilbert (Varambally et al., 2026).

et al., 2024), and its research-level math skills on abstract algebra and category theory with the FATE (Jiang et al., 2026) and LeanCat (Xu et al., 2025) datasets, respectively. We only use the formalized statements from LeanCat and do not include their natural language description. We present a summary of the benchmark results in Table 1.

This minimal prover agent vastly outperforms all the non-agentic Lean 4 theorem provers across all datasets. Additionally, it achieves scores comparable to those of state-of-the-art highly-complex provers, while using a significantly simpler architecture that operates at a fraction of their resources. We provide a thorough cost-performance analysis of this instantiation of AxProverBase with Claude Opus 4.5 in Appendix F, where we show, for instance, that AxProver-Base achieves a similar pass@1 performance to Hilbert in PutnamBench using about 100x less tokens. Out of those, over 70% are input tokens, reducing the cost even further to an average of $12.6 per problem across all the considered datasets (including failed proofs).

Furthermore, the high scores on the FATE and LeanCat datasets indicate that such a system can already be useful in realistic research scenarios, where it can prove a large portion of the theorems involved in complex projects. Additionally, with no dedicated training or fine-tuning, AxProver-Base is readily applicable to different Lean and Mathlib versions.

## 5. Discussion

We have shown that modern foundation models can already become strong formal theorem provers when provided with a simple scaffold that facilitates iterative refinement. With Claude Opus 4.5, a 32k thinking budget and 50 iterations, AxProverBase reaches 54.7% on PutnamBench at pass@1, 98.0% on FATE-M, 66.0% on FATE-H, 24.0% on FATE-X, and 59.0% on LeanCat. All together, these results substantially outperform the baselines based on domain-specific tuning and remain competitive with much more elaborate prover architectures, despite relying on a far simpler design and no dedicated theorem-proving fine-tuning. The results extend beyond competition mathematics, showing that our system is broadly useful across qualitatively different formal domains with practical research applications.

This simple approach not only achieves competitive results, but it also does so rather efficiently. On PutnamBench, AxProverBase is near-parity with Hilbert pass@1 (54.7% vs 55.9%) while operating with substantially different token budgets per problem (4.2M vs 1880.4M). In Appendix F, we show that Hilbert achieves AxProverBase's maximum performance with two orders of magnitude more tokens. On FATE, AxProverBase also shows better token efficiency than all the specialized models, especially considering that most of the tokens in AxProverBase are input tokens, while in those baselines output tokens dominate. The average cost

across the datasets presented in Table 1 has been 12.6$ per sample.

The ablation studies show the role played by each of the elements in our agentic system architecture. Iterative refinement is the main driver for performance, and memory yields the second largest gain. Search tools also improve the performance, although their contribution is smaller than the feedback and memory mechanisms. Our error analysis in Appendix E shows that memory mechanisms and tools reduce code errors and gives the prover more room to explore proof strategies and make steady progress towards a solution. For instance, name and syntax errors drop by 53% and 26%, respectively. Out of the two memory mechanisms, the self-managed context based on self-reflection is especially useful because it can preserve relevant information from past attempts without a fixed time horizon while preventing bloating the context.

At the same time, our experiments show that the agentic framework does not remove the dependence on the underlying model. Indeed, it rather amplifies the differences between them with stronger models benefiting more from the same architecture. For instance, increasing the thinking budget of Claude Opus from 10k to 32k tokens yields the same performance as doubling the number of iterations at significantly lower cost. As frontier LLMs continue to improve, a simple baseline like AxProverBase will improve with them, which makes it a useful moving reference point for separating gains due to better models from gains due to genuine architectural innovations.

Our simple framework allows for flexible experimentation at a relatively low cost, and its modular and extensive design facilitates building upon it. We believe there are many opportunities to improve AxProverBase, as each of its individual components may deserve a dedicated study. Improved versions of the prover could include, for instance, richer retrieval and search tools to support local project context, stronger proof verification using SafeVerify (GasStation-Manager & Lezeau, 2026) or LeanChecker (leanprover contributors, 2026), better memory management systems, or even use fine-tuned models within the same architecture to improve the performance even further. Therefore, AxProverBase provides both a practical prover that is competitive and accessible, as well as a simple and interpretable framework to conduct research on agentic theorem proving.

In conclusion, with AxProverBase we show that a simple theorem-proving agent can already achieve a competitive performance with substantially more elaborate theorem-proving systems with better cost-performance tradeoffs. An LLM embedded in a minimal loop built around compiler feedback, compact context management, and lightweight search tools can make an accessible prover for practical use across multiple domains without any domain-specific fine-tuning. Beyond its practical value, the simplicity and modularity of this design make it a useful reference point for the community: each component can be studied or replaced in isolation, and as frontier models continue to improve, AxProverBase naturally improves with them. Thus, it serves as a strong moving baseline against which future architectural contributions can be measured, providing a clearer separation between gains attributable to architectural innovation and those inherited from stronger foundation models. We provide the code to the full implementation and to reproduce the experiments at https://github.com/Axiomatic-AI/ax-prover-base.

## Acknowledgements

We thank Benjamin Breen and Marco del Tredici for their fruitful discussions.

## Impact Statement

This work introduces AxProverBase, a system that makes AI-assisted theorem proving more effective and accessible. This has potential benefits for accelerating scientific research provided that it may help researchers in mathematics, science, and engineering disciplines adopt formal verification methods into real workflows. More broadly, progress in this direction may contribute to the development of verified scientific reasoning, where AI systems can support not only the generation of hypotheses, derivations, or explanations, but also the production of machine-checkable evidence for their correctness.

These opportunities come with important limitations. A proof assistant certifies a theorem only relative to its formal statement, dependencies, and trusted kernel. However, it does not guarantee that the formal statement faithfully represents the intended scientific claim. Misformalization, benchmark overfitting, or overreliance on automated outputs could therefore lead to misplaced confidence. We view such systems as tools for augmenting expert researchers, and emphasize open-source implementation, reproducibility, transparent evaluation, and human oversight as important safeguards.

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

# A. PutnamBench Leaderboard

Table 2 shows the Lean section of the PutnamBench leaderboard as of January 2026. The benchmark consists of 672 problems from Putnam competitions. This table reports the number of problems solved and the computational budget used by each system.

*Table 2.* PutnamBench Lean Leaderboard (out of 672 problems)

| Model | Reference | Solved | Compute Budget |
|---|---|---|---|
| Aleph Prover ($1400 limit) | (Logical Intelligence, 2026) | 668 | Avg $68 |
| Aleph Prover ($400 limit) | (Logical Intelligence, 2026) | 637 | Avg $54 |
| Seed-Prover 1.5 (ByteDance) | (Chen et al., 2025a) | 581 | 10 H20 days/problem |
| Aleph Prover ($100 limit) | (Logical Intelligence, 2026) | 500 | Avg $23 |
| Hilbert | (Varambally et al., 2026) | 462 | avg pass@1840 |
| Seed-Prover (ByteDance) | (Chen et al., 2025b) | 329 | MEDIUM |
| Ax-Prover | (Breen et al., 2025) | 91 | pass@1, avg. 100 tool calls |
| Goedel-Prover-V2 | (Lin et al., 2026) | 86 | pass@184 |
| DeepSeek-Prover-V2 | (Ren et al., 2025) | 47 | pass@1024 |
| GPT-5 (ReAct, 10 turns) | – | 28 | pass@1, 10 tool calls |
| DSP+ | (Cao et al., 2025) | 23 | pass@128 |
| Bourbaki | (Zimmer et al., 2025) | 14 | pass@512 |
| Kimina-Prover-7B-Distill | (Wang et al., 2025a) | 10 | pass@192 |
| Self-play Theorem Prover | (Dong & Ma, 2025) | 8 | pass@3200 |
| ABEL | (Gloeckle et al., 2024) | 7 | pass@596 |
| Goedel-Prover-SFT | (Lin et al., 2025) | 7 | pass@512 |
| InternLM2.5-StepProver | (Wu et al., 2025) | 6 | pass@2x32x600 |
| InternLM 7B | (Cai et al., 2024) | 4 | pass@4096 |
| gemini-2.5-pro-exp-0325 | (Comanici et al., 2025) | 3 | pass@1 |
| gemini-2.0-flash-thinking-121 | – | 1 | pass@1 |
| Deepseek R1 | (Guo et al., 2025) | 1 | pass@1 |
| COPRA (GPT-4o) | (Thakur et al., 2024) | 1 | pass@1 |
| GPT-4o | (OpenAI et al., 2024) | 1 | pass@10 |

# B. Prompts

This section contains the prompts used in our experiments. Every block in the agent architecture described in Section 3 has a system prompt, and then it receives a user prompt with specific information about the task at hand. In Appendix C, we provide explicit prompt examples.

## B.1. Proposer prompt

We change the system prompt of the proposer depending on whether it should follow an iterative or single-shot proof strategy. Then, we compose a user prompt by combining the "proposer user prompt", that provides the original Lean file content and specifies the target theorem (see below), together with information about its most recent attempt (if available), and additional context provided by the memory mechanism (if available).

---

**System Prompt Iterative Proof**

```
You are an LLM acting as a Lean 4 proof expert in an ITERATIVE proof
development process.

## Your Role

Your task is to propose the next iteration of a Lean 4 proof
(Lean version 4.24). Your code will be compiled with `lake build`,
```

and the results (successes, errors, remaining goals) will inform
the next iteration. Think carefully about the current proof state
and the knowledge gathered from the previous attempt to make as much
progress as possible with this new attempt.

**CRITICAL: While you do NOT need to produce a complete working proof
in a single attempt, you DO NEED to submit a final full proof regardless
of the difficulty and the time it takes to complete it.
NEVER give up, if something is complex, break it down into manageable
steps, sketch out the proof structure, and work relentlessly
to solve it. You can prove everything that will be given to you.**

## Iteration Strategy

- **Incomplete proofs can help**: You may use incomplete proofs to
  strategically explore proof structure and understand what goals
  need to be proven. However, they are not a valid final result
- **How `sorry` is handled**: `sorry` statements are replaced with
  empty strings before compilation, causing Lean to report the
  specific unsolved goals at those locations. This is intentional:
  these error messages provide detailed feedback about what remains
  to be proven.
- **Build feedback is information**: Compilation errors and goal states
  help you understand what to try next
- **Experience is valuable**: We keep track of the main lessons learned
  across all previous attempts. Use this information to avoid repeating
  past mistakes and evaluate the current appraoch.
- **Make steady progress**: Each iteration counts! Therefore, you
  should aim to fix existing errors, complete missing parts of the
  proof, or even explore a new approach if the current one is clearly
  not working.
- **First attempts**: Try a promising approach and see what feedback
  you get
- **Later iterations**: Analyze the feedback from the previous build
  and address the issues directly

<requirements>
1. **Preserve Theorem Statement**
    - NEVER modify the theorem signature, name or type
    - Only change the proof body (after :=)
    - Work ONLY on the target theorem

2. **Proof Development Strategy**
    - Use appropriate Lean 4 tactics to make progress
    - Leverage the build feedback and experience to inform the next
      action and make steady progress towards the final proof
    - The proof must eventually be complete and fully build without
      any `sorry` statements

3. **Quality Standards**
    - Use appropriate Lean 4 tactics and syntax
    - Reference Mathlib lemmas when applicable
    - Add imports as needed for any lemmas or tactics you use
</requirements>

<output-format>
Return structured output with these fields.
In all cases, please produce all the fields in your output:

**reasoning**:
Think carefully about the theorem at hand, analyze the state of the
proof together with the experience and the build feedback from the
last attempt, evaluate the most promising approaches, and devise a

```
suitable proof strategy.
Identify what went wrong and find a suitable strategy to address it.
Capitalize on the previous experience and respect the lessons learned
from the past to avoid repeating mistakes.

**imports**: Required imports NOT already in the file
- Format: ["Mathlib.Topology.Basic", "MyProject.Algebra.Ring"]
- Only include what's necessary

**opens**: Namespaces to open NOT already opened
- Format: ["Algebra", "MeasureTheory"]
- Only include what's necessary

**updated_theorem**: Updated code for theorem
- Theorem statement and proof body
- It only contains the specific theorem that was changed
</output-format>

<examples>
Target theorem: `theorem add_zero (n : Nat) : n + 0 = n := sorry`
<example>
**First attempt - exploring tactics**

imports: []
opens: []
updated_theorem:
'''lean
theorem add_zero (n : Nat) : n + 0 = n := by
  simp
'''

(Next iteration would receive feedback about whether `simp` solved it
or what goals remain)
</example>

<example>
**Second iteration - incomplete proof with unsolved goals**
The code from the previous iteration builds successfully but goals
remain: `|- n + 0 = n`. Therefore, the `simp` tactic alone doesn't
solve this goal. A possible strategy now is to use induction to break
this into base and inductive cases, leaving sorries to see what
specific goals need to be proven in each case.

imports: []
opens: []
updated_theorem:
'''lean
theorem add_zero (n : Nat) : n + 0 = n := by
  induction n with
  | zero => sorry
  | succ n ih => sorry
'''

(This intentionally leaves sorries to get feedback about the base and
inductive case goals)
</example>

<example>
**Third iteration - Closing the goals**
Sorry #1 at line 3, column 13 is |- 0 + 0 = 0, and sorry #2 at line 4,
column 13 is \|- n + 1 + 0 = n + 1. Without these sorries, we have
inspected the goal state in multiple points of the proof.
Now we can proceed to close these goals.
```

```
imports: ["Mathlib.Data.Nat.Basic"]
opens: []
updated_theorem:
```lean
theorem add_zero_1 (n : Nat) : n + 0 = n := by
  induction n with
  | zero => rfl
  | succ n ih => rw [Nat.add_succ, ih]
```
</example>

<example>
**Fourth iteration - fixing specific error**
The previous attempt's build failed with error at line 4: 'unknown
identifier Nat.add_succ'. The proof attempted to use a lemma that
doesn't exist or isn't imported (the correct name is `Nat.succ_add`).
A reasonable next step is to search for the correct lemma name,
although, after looking at it carefully, this is a case that
`omega` (specialized for integer and natural linear arithmetic
problems) can solve.

imports: []
opens: []
updated_theorem:
```lean
theorem add_zero (n : Nat) : n + 0 = n := by
  omega
```
</example>
</examples>
```

## System Prompt Single-Shot Proof

```
You are an LLM acting as a Lean 4 proof expert in a SINGLE-SHOT process.

# Your Role
Your task is to propose a complete and valid Lean 4 proof
(Lean version 4.24).
Your code will be checked with `lake build`, although you will not be
able to see the build result nor receive any feedback.
You can use the tools provided to you to help you with your task.

**CRITICAL: You MUST produce a complete proof in this single attempt
without any placeholder (like `sorry` or `admit`). You DO NEED to
submit a final full proof regardless of the difficulty and the time
it takes to complete it. NEVER give up, if something is complex, break
it down into manageable steps, and think about the proof structure
before writing it. You can prove everything that will be given to
you.**

<requirements>
1. **Preserve Theorem Statement**
    - NEVER modify the theorem signature, name or type
    - Only change the proof body (after :=)
    - Work ONLY on the target theorem

2. **Proof Development Strategy**
    - Use appropriate Lean 4 tactics to make progress
    - The proof must be complete and valid lean4 proof and fully
      successfully
      without any `sorry` or `admit` statements, nor introducing new
      axioms.
```

```
3. **Quality Standards**
    - Use appropriate Lean 4 tactics and syntax
    - Reference Mathlib lemmas when applicable
    - Add imports as needed for any lemmas or tactics you use
</requirements>

<output-format>
Return structured output with these fields.
In all cases, please produce all the fields in your output:

**reasoning**:
Think carefully about the theorem at hand, evaluate the most promising
approaches to prove it, and devise a suitable proof strategy.

**imports**: Required imports NOT already in the file
- Format: ["Mathlib.Topology.Basic", "MyProject.Algebra.Ring"]
- Only include what's necessary

**opens**: Namespaces to open NOT already opened
- Format: ["Algebra", "MeasureTheory"]
- Only include what's necessary

**updated_theorem**: Updated code for theorem
- Theorem statement and proof body
- It only contains the specific theorem that was changed
</output-format>

<examples>
<example1>
imports: []
opens: []
updated_theorem:
```lean
theorem lt_or_gt_zero_if_neq_zero {x : Real} (h : x != 0) :
    x < 0 \/ x > 0 := by
  rcases lt_trichotomy x 0 with xlt | xeq | xgt
  · left
    exact xlt
  · contradiction
  · right; exact xgt
```
</example1>

<example2>
imports: [Mathlib.Tactic]
opens: []
updated_theorem:
```lean
theorem finite_series_eq {n:Nat} {Y:Type*} (X: Finset Y) (f: Y -> Real)
  (g: Icc (1:Int) n -> X) (hg: Function.Bijective g) :
    Sum i in X, f i = Sum i in Icc (1:Int) n,
      (if hi:i in Icc (1:Int) n then f (g < i, hi >) else 0) := by
  symm
  convert sum_bij (t:=X) (fun i hi |-> g < i, hi > ) _ _ _ _
  . aesop
  . intro _ _ _ _ h; simpa [Subtype.val_inj, hg.injective.eq_iff]
      using h
  . intro b hb; have := hg.surjective < b, hb >; grind
  intros; simp_all
```
</example2>

<example3>
```

```
imports: [Mathlib.Tactic]
opens: []
updated_theorem:
```lean
theorem EqualCard.univ (X : Type) : EqualCard (.univ : Set X) X :=
  < Subtype.val, Subtype.val_injective, by intro _; aesop >
```
</example3>
</examples>
```

## Proposer User Prompt

```
Complete the proof for the following target theorem within the given
file.

<target>
{target theorem identifier}
</target>

<complete-file>
```lean
{complete Lean file content}
```
</complete-file>
```

## Previous Attempt User Prompt

```
This is your previous attempt at proving the theorem, containing your
previous reasoning, the proposed code, and the feedback that you
received from either building the code or a reviewer agent.

<attempt>
<reasoning>
{reasoning}
</reasoning>


{code}


<feedback>
{feedback}
</feedback>
</attempt>
```

## Proposer Self-Managed-Context User Prompt

```
What follows is additional context with relevant lessons learned from
your previous attempts at proving this theorem.

<experience>
{experience generated by the memory module}
</experience>
```

## Proposer History of Past Attempts User Prompt

```
These are your most recent attempts preceding the current one (right
above) in reverse chronological order (most recent first):

<previous-attempts>
{list of previous attempts with reasoning, code and feedback each}
</previous-attempts>
```

## B.2. Memory prompt

We only need prompts in the memory module for the case where we use a self-managed context.

---

**Context Summary System Prompt**

```
An LLM agent is trying to prove a theorem in Lean 4 and it failed to
successfully complete the full proof.
This prover agent can work iteratively and take multiple attempts to
prove the theorem, but it does not have access to its own history of
previous attempts. Instead, it is your task to provide it with
meaningful context condensing the experience gained from its previous
attempts, which will help it succeed in its next one at proving the
theorem. Therefore, when the prover takes another attempt at proving
the theorem, it will be given both the current state of the proof
(code and feedback) plus the context that you prepare.

You will recieve a message containing the last attempt's information:
reasoning that the LLM output before generating the code, the Lean 4
code itself, and the feedback that comes either from the `lake build`
output or a reviewer agent. Additionally, the message will contain
the previous context that you had prepared for the prover agent
synthesizing its past experience to help it generate this code. If the
previous context was empty, it means this was the first attempt at
proving the theorem.

Reflect on the current attempt in light of the previous context and
experience, and extract the key lessons to avoid repeating the same
mistakes and any of the previous ones in the future.
Write the context that will be given to the prover agent in the next
iteration. Ensure that the important information from the experience
captured in the previous context is never lost when writing the new
context, as we want to prevent the prover agent from repeating the
same mistakes even after multiple future iterations.
You should stick to the facts and refrain from proposing new actions
or strategies, as the prover agent will draw the necessary conclusions
by itself with the information that you provide.
Keep in mind that this context should not be excessively long, and it
should be concise and to the point with the goal of maximizing the
performance of the prover agent.

Your output will be given to the prover agent in another message,
where it will be the content embedded within the <experience> tag
in the following template:
<message-template>
What follows is additional context with relevant lessons learned
from your previous attempts at proving this theorem.

<experience>
{experience}
</experience>
</message-template>
```

---

**Context Summary User Prompt**

```
<attempt>
<reasoning>
{reasoning}
</reasoning>


{code}

```

```
<feedback>
{feedback}
</feedback>
</attempt>

<previous-context>
{previous_context}
</previous-context>
```

## B.3. Reviewer prompt

---

**Reviewer System Prompt**

```
You are an LLM acting as a Lean 4 expert checking proofs. You focus on
three things:

<check-1>
**Statement Preserved?**
Verify the theorem statement wasn't modified. True if identical, False
if changed.
</check-1>

<check-2>
**No Sorry in NEW PROOF Body?**
Search for "sorry"/"admit" in the PROPOSED body only (ORIGINAL has
sorry - that'sexpected).
True if no sorry/admit, False if found.
</check-2>

<check-3>
**No Other Issues?**
Note any other issues (undefined refs, syntax errors, logic problems).
True if no issues, False if problems found.
</check-3>

<examples>
<example>
**Typical case**
```lean
-- ORIGINAL
theorem foo (n : Nat) : n + 0 = n := sorry

-- NEW PROOF
theorem foo (n : Nat) : n + 0 = n := by
  have h1 := foo_h1
  exact h1
```
check1: True, check2: True, check3: False, approved: False
reasoning: "Statement preserved, no sorry, but references undefined
foo_h1"
</example>

<example>
**Statement changed**
```lean
-- ORIGINAL
theorem foo (n : Nat) : n + 0 = n := sorry

-- NEW PROOF
theorem foo (n m : Nat) : n + m = n := by ...
```
check1: False, check2: True, check3: True, approved: False
reasoning: "Statement changed - added parameter m"
```

```
</example>

<example>
**Has sorry**
```lean
-- ORIGINAL
theorem foo (n : Nat) : n + 0 = n := sorry

-- NEW PROOF
theorem foo (n : Nat) : n + 0 = n := by sorry
```
check1: True, check2: False, check3: True, approved: False
reasoning: "Statement preserved but proof still contains sorry
</example>
</examples>
```

**Reviewer User Prompt**

```
<original>
```lean
{original_theorem}
```
</original>

<proposed>
-- NEW PROOF
```lean
{proposed_proof}
```
</proposed>
```

## C. Example: complete proof process

Here, we provide a full example of the proof generation process, providing all the inputs and outputs of the main architecture components described in Section 3. We omit the system prompts provided in Appendix B for simplicity, given that they are fixed.

We take problem 116 from the FATE-M dataset (Jiang et al., 2026), which asks to prove a simple identity for a non-commutative ring. We have chosen this problem because it is simple enough to be understood by a broad audience, yet not sufficiently straightforward for our agent to prove in its first attempt, requiring a second iteration to finish off the proof. Additionally, we want to provide a real example that has been used in the evaluation of the agent reported in Section 4.3.

### C.1. First iteration

The proof process starts with the proposer. We give it the system prompt for an iterative proof strategy detailed in Appendix B.1, which we omit here.

**Proposer**

**User Input**

```
Complete the proof for the following target theorem within the given
file.
<target>
LeanBenchmarks.FATE_M.116:pow_add_pow_eq
</target>

<complete-file>
```lean
```

```
import Mathlib

theorem pow_add_pow_eq {R : Type*} (a b : R) [Ring R]
    (h : a * b = - (b * a)) :
    (a + b) ^ 2 = a ^ 2 + b ^ 2 ∧ (a - b) ^ 2 = a ^ 2 + b ^ 2 := by
      sorry
```
</complete-file>

**Agent reasoning**

```
Let me analyze this theorem. We need to prove that if
$a * b = -(b * a)$, then:
1. $(a + b)^2 = a^2 + b^2$
2. $(a - b)^2 = a^2 + b^2$

In a ring, $(a + b)^2 = a^2 + ab + ba + b^2$. Given that $ab = -ba$,
we have $ab + ba = ab - ab = 0$. So $(a + b)^2 = a^2 + b^2$.

Similarly, $(a - b)^2 = a^2 - ab - ba + b^2 = a^2 - ab + ab + b^2
= a^2 + b^2$ (since $-ba = ab$).

Let me think about how to prove this in Lean 4. I'll need to expand
the squares and use the hypothesis.

Let me search for relevant lemmas about ring operations and powers.
```

**Parallel tool calls**

**Tool:** LeanSearch
**Query:** add_sq ring power two expansion
**Result:**

```
=== add_sq ring power two expansion (6 matches) ===
```
• PowerSeries.exp_mul_exp_eq_exp_add [theorem]
  [Algebra ℚ A] (a b : A) : rescale a (exp A) * rescale b (exp A)
  = rescale (a + b) (exp A)
  Doc: Shows that $e^{aX} * e^{bX} = e^{(a + b)X}$
• TrivSqZeroExt.fst_add [theorem]
  [Add R] [Add M] (x₁ x₂ : tsze R M) : (x₁ + x₂).fst = x₁.fst + x₂.fst
• PowerSeries.instAddGroup [instance]
  [AddGroup R] : AddGroup R[[X]]
• TrivSqZeroExt.add [instance]
  [Add R] [Add M] : Add (tsze R M)
• add_mul_self_eq [theorem]
  (a b : α) : (a + b) * (a + b) = a * a + 2 * a * b + b * b
• PowerSeries.coe_add [theorem]
  : ((f + g : R[[X]]) : R(X)) = f + g
```

**Tool:** LeanSearch
**Query:** sub_sq ring power two expansion
**Result:**

```
=== sub_sq ring power two expansion (6 matches) ===
```
• Subring.coe_pow [theorem]
  (x : s) (n : ℕ) : ↑(x ^ n) = (x : R) ^ n
• polynomial_expand_eq [theorem]
  (f : R[X]) : expand R p f = (f.map (frobeniusEquiv R p).symm) ^ p
• Subring.pow_mem [theorem]
  {x : R} (hx : x ∈ s) (n : ℕ) : x ^ n ∈ s
• Polynomial.expand_pow [theorem]
  (f : R[X]) : expand R (p ^ q) f = (expand R p)^[q] f

23
```

- FiniteField.expand_card [theorem]
  (f : K[X]) : expand K q f = f ^ q
- PowerSeries.instSubsingleton [instance]
  [Subsingleton R] : Subsingleton R[[X]]

---

**Agent reasoning**

```
Let me analyze the theorem. I need to prove that
if `a * b = -(b * a)`, then:
1. `(a + b)^2 = a^2 + b^2`
2. `(a - b)^2 = a^2 + b^2`

I'll use ring expansion. Let me think about the proof:

For (a + b)^2:
- (a + b)^2 = a^2 + a*b + b*a + b^2
- Since a*b = -(b*a), we have a*b + b*a = -(b*a) + b*a = 0
- So (a + b)^2 = a^2 + b^2

For (a - b)^2:
- (a - b)^2 = a^2 - a*b - b*a + b^2
- Since a*b = -(b*a), we have -a*b - b*a = -a*b + a*b = 0
- So (a - b)^2 = a^2 + b^2

I'll try to prove this using `ring_nf` or similar tactics, combined
with rewriting using the hypothesis. Let me construct the proof.

The key insight is that from `h : a * b = -(b * a)`, we can derive
`a * b + b * a = 0`.

Let me write out the proof step by step:
```

---

**Proposed proof**

```
theorem pow_add_pow_eq {R : Type*} (a b : R) [Ring R]
    (h : a * b = - (b * a)) :
    (a + b) ^ 2 = a ^ 2 + b ^ 2 ∧ (a - b) ^ 2 = a ^ 2 + b ^ 2 := by
  have hab : a * b + b * a = 0 := by
    rw [h]
    ring
  constructor
  · ring_nf
    rw [add_comm (a * b) (b * a), hab]
    ring
  · ring_nf
    rw [add_comm (a * b) (b * a), hab]
    ring
```

We update the proposed updated theorem in a temporary copy of the target file, which the reviewer tries to compile. In this case, the code does not compile.

**Reviewer**

**Compilation output**

```
× [7351/7351] Building LeanBenchmarks.FATE_M.tmp_116_ul11ixq2 (6.6s)

+- Error at line 8:4
|      ring
|      ^^^
+- ring_nf made no progress on goal

+- Error at line 11:8
|      rw [add_comm (a * b) (b * a), hab]
|         ^^^
+- Tactic `rewrite` failed: Did not find an occurrence of the pattern
  a * b + b * a
in the target expression
  (a + b) ^ 2 = a ^ 2 + b ^ 2

case left
R : Type u_1
a b : R
inst† : Ring R
h : a * b = -(b * a)
hab : a * b + b * a = 0
⊢ (a + b) ^ 2 = a ^ 2 + b ^ 2

+- Error at line 14:8
|      rw [add_comm (a * b) (b * a), hab]
|         ^^^
+- Tactic `rewrite` failed: Did not find an occurrence of the pattern
  a * b + b * a
in the target expression
  (a - b) ^ 2 = a ^ 2 + b ^ 2

case right
R : Type u_1
a b : R
inst† : Ring R
h : a * b = -(b * a)
hab : a * b + b * a = 0
⊢ (a - b) ^ 2 = a ^ 2 + b ^ 2

error: Lean exited with code 1
Some required targets logged failures:
- LeanBenchmarks.FATE_M.tmp_116_ul11ixq2
error: build failed
```

The information of the attempt is given to the memory processor. In this case, we use the self-managed context for the memory whose system prompt is specified in Appendix B.2.

**Memory**

**User Input**

```
<attempt>
<reasoning>
Let me analyze the theorem. I need to prove that
if `a * b = -(b * a)`, then:
1. `(a + b)^2 = a^2 + b^2`
2. `(a - b)^2 = a^2 + b^2`
```

```
I'll use ring expansion. Let me think about the proof:

For (a + b)^2:
- (a + b)^2 = a^2 + a*b + b*a + b^2
- Since a*b = -(b*a), we have a*b + b*a = -(b*a) + b*a = 0
- So (a + b)^2 = a^2 + b^2

For (a - b)^2:
- (a - b)^2 = a^2 - a*b - b*a + b^2
- Since a*b = -(b*a), we have -a*b - b*a = -a*b + a*b = 0
- So (a - b)^2 = a^2 + b^2

I'll try to prove this using `ring_nf` or similar tactics, combined
with rewriting using the hypothesis. Let me construct the proof.

The key insight is that from `h : a * b = -(b * a)`, we can derive
`a * b + b * a = 0`.

Let me write out the proof step by step:
</reasoning>


theorem pow_add_pow_eq {R : Type*} (a b : R) [Ring R]
    (h : a * b = - (b * a)) :
    (a + b) ^ 2 = a ^ 2 + b ^ 2 ∧ (a - b) ^ 2 = a ^ 2 + b ^ 2 := by
  have hab : a * b + b * a = 0 := by
    rw [h]
    ring
  constructor
  · ring_nf
    rw [add_comm (a * b) (b * a), hab]
    ring
  · ring_nf
    rw [add_comm (a * b) (b * a), hab]
    ring


<feedback>
BUILD FAILED:

× [7351/7351] Building LeanBenchmarks.FATE_M.tmp_116_ul11ixq2 (6.6s)

+- Error at line 8:4
|      ring
|      ^^^
+- ring_nf made no progress on goal

+- Error at line 11:8
|      rw [add_comm (a * b) (b * a), hab]
|         ^^^
+- Tactic `rewrite` failed: Did not find an occurrence of the pattern
  a * b + b * a
in the target expression
  (a + b) ^ 2 = a ^ 2 + b ^ 2

case left
R : Type u_1
a b : R
inst† : Ring R
h : a * b = -(b * a)
hab : a * b + b * a = 0
⊢ (a + b) ^ 2 = a ^ 2 + b ^ 2
```

```
+- Error at line 14:8
|       rw [add_comm (a * b) (b * a), hab]
|          ^^^
+- Tactic `rewrite` failed: Did not find an occurrence of the pattern
  a * b + b * a
in the target expression
  (a - b) ^ 2 = a ^ 2 + b ^ 2

case right
R : Type u_1
a b : R
inst† : Ring R
h : a * b = -(b * a)
hab : a * b + b * a = 0
⊢ (a - b) ^ 2 = a ^ 2 + b ^ 2

error: Lean exited with code 1
Some required targets logged failures:
- LeanBenchmarks.FATE_M.tmp_116_ul11ixq2
error: build failed
</feedback>
</attempt>

<previous-context>

</previous-context>
```

**Agent reasoning**

Looking at this first attempt, the prover is trying to prove a theorem about anticommutative multiplication in a ring. The key insight that `a * b + b * a = 0` is correct, but the approach has issues.

Problems identified:
1. The `ring_nf` tactic doesn't help here because the ring is non-commutative (it's a general Ring, not a CommRing), so `ring` and `ring_nf` tactics won't work as expected for simplification.
2. After `ring_nf`, the goal remains `(a + b) ^ 2 = a ^ 2 + b ^ 2`, so the rewrite fails because the pattern `a * b + b * a` doesn't appear in the unexpanded form.

The issue is that in a non-commutative ring, the `ring` tactic assumes commutativity and won't help. The prover needs to manually expand the squares and use the hypothesis directly.

Key lessons:
1. `ring` and `ring_nf` tactics assume commutativity, so they don't work properly for general (non-commutative) rings.
2. The prover needs to manually expand `(a + b)^2` and `(a - b)^2` and then simplify using the hypothesis.
3. Square expansion: `x^2 = x * x`, then
   `(a+b)*(a+b) = a*a + a*b + b*a + b*b`
4. Need to use lemmas like `add_mul`, `mul_add`, `pow_two` or `sq` to expand, then use the hypothesis that `a*b = -(b*a)` to simplify `a*b + b*a = 0`.

Let me write a concise context for the prover.

**Experience context**

```
Key lessons from this attempt:

1. **`ring` and `ring_nf` don't work for non-commutative rings**:
   The theorem is about a general `Ring R`, which is non-commutative.
   The `ring` family of tactics assumes commutativity, so they
   cannot    simplify expressions involving `a * b` and `b * a` as
   separate terms.

2. **Manual expansion is required**: Since `ring_nf` doesn't expand
   the goal, the pattern `a * b + b * a` never appears in the
   expression `(a + b) ^ 2 = a ^ 2 + b ^ 2`, causing the rewrite to
   fail.

3. **Correct approach**: Need to manually expand the squares using
   lemmas like:
   - `pow_two x` or `sq x` to rewrite `x^2` as `x * x`
   - `add_mul`, `mul_add` to distribute multiplication over addition
   - `sub_mul`, `mul_sub` for the subtraction case

4. **The helper lemma `hab : a * b + b * a = 0` is correct** and will
   be useful once the expressions are properly expanded.
   From `h : a * b = -(b * a)`,
   we get `a * b + b * a = -(b * a) + b * a = 0`.

5. **After full expansion**, `(a + b)^2`
   becomes `a*a + a*b + b*a + b*b`, which equals
   `a^2 + (a*b + b*a) + b^2 = a^2 + 0 + b^2 = a^2 + b^2`.
```

Notice that the previous context is empty in the user input. This is because this is the first iteration, so there is no context aggregating past experience yet. In the following iteration, this agent's output would fill that gap.

## C.2. Second iteration

Given that the agent has not managed to successfully prove the theorem, it goes on for another iteration. Now, it will be able to leverage the learning from the first iteration to produce a better proof candidate.

**Proposer**

**User Input**

```
Complete the proof for the following target theorem within the given
file.
<target>
LeanBenchmarks.FATE_M.116:pow_add_pow_eq
</target>

<complete-file>
```lean
import Mathlib

theorem pow_add_pow_eq {R : Type*} (a b : R) [Ring R]
    (h : a * b = - (b * a)) :
    (a + b) ^ 2 = a ^ 2 + b ^ 2 ∧ (a - b) ^ 2 = a ^ 2 + b ^ 2 := by
      sorry
```
</complete-file>
```

This is your previous attempt at proving the theorem, containing your
previous reasoning, the proposed code, and the feedback that you
received from either building the code or a reviewer agent.

```
<attempt>
<reasoning>
Let me analyze the theorem. I need to prove that
if `a * b = -(b * a)`, then:
1. `(a + b)^2 = a^2 + b^2`
2. `(a - b)^2 = a^2 + b^2`

I'll use ring expansion. Let me think about the proof:

For (a + b)^2:
- (a + b)^2 = a^2 + a*b + b*a + b^2
- Since a*b = -(b*a), we have a*b + b*a = -(b*a) + b*a = 0
- So (a + b)^2 = a^2 + b^2

For (a - b)^2:
- (a - b)^2 = a^2 - a*b - b*a + b^2
- Since a*b = -(b*a), we have -a*b - b*a = -a*b + a*b = 0
- So (a - b)^2 = a^2 + b^2

I'll try to prove this using `ring_nf` or similar tactics, combined
with rewriting using the hypothesis. Let me construct the proof.

The key insight is that from `h : a * b = -(b * a)`, we can derive
`a * b + b * a = 0`.

Let me write out the proof step by step:
</reasoning>


theorem pow_add_pow_eq {R : Type*} (a b : R) [Ring R]
    (h : a * b = - (b * a)) :
    (a + b) ^ 2 = a ^ 2 + b ^ 2 ∧ (a - b) ^ 2 = a ^ 2 + b ^ 2 := by
  have hab : a * b + b * a = 0 := by
    rw [h]
    ring
  constructor
  · ring_nf
    rw [add_comm (a * b) (b * a), hab]
    ring
  · ring_nf
    rw [add_comm (a * b) (b * a), hab]
    ring


<feedback>
BUILD FAILED:

✗ [7351/7351] Building LeanBenchmarks.FATE_M.tmp_116_ul11ixq2 (6.6s)

+- Error at line 8:4
|      ring
|      ^^^
+- ring_nf made no progress on goal

+- Error at line 11:8
|      rw [add_comm (a * b) (b * a), hab]
|         ^^^
+- Tactic `rewrite` failed: Did not find an occurrence of the pattern
  a * b + b * a
```

```
in the target expression
  (a + b) ^ 2 = a ^ 2 + b ^ 2

case left
R : Type u_1
a b : R
inst† : Ring R
h : a * b = -(b * a)
hab : a * b + b * a = 0
⊢ (a + b) ^ 2 = a ^ 2 + b ^ 2

+- Error at line 14:8
|      rw [add_comm (a * b) (b * a), hab]
|          ^^^
+- Tactic `rewrite` failed: Did not find an occurrence of the pattern
  a * b + b * a
in the target expression
  (a - b) ^ 2 = a ^ 2 + b ^ 2

case right
R : Type u_1
a b : R
inst† : Ring R
h : a * b = -(b * a)
hab : a * b + b * a = 0
⊢ (a - b) ^ 2 = a ^ 2 + b ^ 2

error: Lean exited with code 1
Some required targets logged failures:
- LeanBenchmarks.FATE_M.tmp_116_ul11ixq2
error: build failed
</feedback>
</attempt>
```

What follows is additional context with relevant lessons learned
from your previous attempts at proving this theorem.

```
<experience>
Key lessons from this attempt:
```

1. **`ring` and `ring_nf` don't work for non-commutative rings**:
   The theorem is about a general `Ring R`, which is non-commutative.
   The `ring` family of tactics assumes commutativity, so they
   cannot    simplify expressions involving `a * b` and `b * a` as
   separate terms.

2. **Manual expansion is required**: Since `ring_nf` doesn't expand
   the goal, the pattern `a * b + b * a` never appears in the
   expression `(a + b) ^ 2 = a ^ 2 + b ^ 2`, causing the rewrite to
   fail.

3. **Correct approach**: Need to manually expand the squares using
   lemmas like:
   - `pow_two x` or `sq x` to rewrite `x^2` as `x * x`
   - `add_mul`, `mul_add` to distribute multiplication over addition
   - `sub_mul`, `mul_sub` for the subtraction case

4. **The helper lemma `hab : a * b + b * a = 0` is correct** and will
   be useful once the expressions are properly expanded.
   From `h : a * b = -(b * a)`,
   we get `a * b + b * a = -(b * a) + b * a = 0`.

```
5. **After full expansion**, `(a + b)^2`
   becomes `a*a + a*b + b*a + b*b`, which equals
   `a^2 + (a*b + b*a) + b^2 = a^2 + 0 + b^2 = a^2 + b^2`.
</experience>
```

## Agent reasoning

Let me analyze the problem more carefully. The feedback tells me
that `ring` and `ring_nf` don't work because the ring is non-
commutative. I need to manually expand the squares using basic lemmas.

Let me think about the approach:

1. First, I'll prove the helper lemma `hab : a * b + b * a = 0`
from `h : a * b = -(b * a)`.

2. For `(a + b)^2 = a^2 + b^2`:
   - Expand `(a + b)^2` = `(a + b) * (a + b)`
   - = `a * (a + b) + b * (a + b)` (using `add_mul`)
   - = `a * a + a * b + b * a + b * b` (using `mul_add`)
   - = `a^2 + (a * b + b * a) + b^2`
   - = `a^2 + 0 + b^2` (using `hab`)
   - = `a^2 + b^2`

3. Similarly for `(a - b)^2 = a^2 + b^2`:
   - Expand `(a - b)^2` = `(a - b) * (a - b)`
   - = `a * (a - b) - b * (a - b)` (using `sub_mul`)
   - = `a * a - a * b - b * a + b * b` (using `mul_sub`)
   - = `a^2 - (a * b + b * a) + b^2`
   - = `a^2 - 0 + b^2` (using `hab`)
   - = `a^2 + b^2`

Let me search for the right lemmas to use.

Actually, let me think about this more carefully.

I need to identify the key Mathlib lemmas: `sq` or `pow_two` to
convert between `x^2` and `x * x`, along with `add_mul`, `mul_add`,
`sub_mul`, and `mul_sub` for distributing multiplication over
addition and subtraction. These will let me expand the squared terms
systematically and then apply the anticommutativity condition to
simplify.

## Parallel tool calls

**Tool:** LeanSearch
**Query:** pow_two multiplication expansion
**Result:**

```
=== pow_two multiplication expansion (6 matches) ===
• BitVec.twoPow_mul_twoPow_eq [theorem]
  {w : Nat} (i j : Nat) : twoPow w i * twoPow w j = twoPow w (i + j)
  Doc: 2^i * 2^j = 2^(i + j) with bitvectors as well
• Polynomial.expand_pow [theorem]
  (f : R[X]) : expand R (p ^ q) f = (expand R p)^[q] f
• BitVec.mul_twoPow_eq_shiftLeft [theorem]
  (x : BitVec w) (i : Nat) : x * (twoPow w i) = x <<< i
• BitVec.twoPow_mul_eq_shiftLeft [theorem]
  (x : BitVec w) (i : Nat) : (twoPow w i) * x = x <<< i
• MvPolynomial.expand_zmod [theorem]
```

```
     (f : MvPolynomial σ (ZMod p)) : expand p f = f ^ p
● BitVec.twoPow [definition]
    (w : Nat) (i : Nat) : BitVec w
  Doc: `twoPow w i` is the bitvector `2^i` if `i < w`, and `0`
  otherwise. In other words, it is 2 to the
power `i`.
From the bitwise point of view, it has the `i`th bit as `1` and all
other bits as `0`.
```

**Tool:** LeanSearch
**Query:** add_mul mul_add distributive ring
**Result:**

```
=== add_mul mul_add distributive ring (6 matches) ===
● AddHom.mulLeft [definition]
    [Distrib R] (r : R) : AddHom R R
  Doc: Left multiplication by an element of a type with distributive
  multiplication is an `AddHom`.
● Mathlib.Tactic.Ring.add_mul [theorem]
    {d : R} (_ : (a₁ : R) * b = c₁) (_ : a₂ * b = c₂)
    (_ : c₁ + c₂ = d) : (a₁ + a₂) * b = d
● AddHom.mulRight [definition]
    [Distrib R] (r : R) : AddHom R R
  Doc: Left multiplication by an element of a type with distributive
  multiplication is an `AddHom`.
● Mathlib.Tactic.Ring.mul_add [theorem]
    {d : R} (_ : (a : R) * b₁ = c₁) (_ : a * b₂ = c₂)
    (_ : c₁ + 0 + c₂ = d) : a * (b₁ + b₂) = d
● smul_add [theorem]
    (a : M) (b₁ b₂ : A) : a • (b₁ + b₂) = a • b₁ + a • b₂
● add_smul [theorem]
    : (r + s) • x = r • x + s • x
```

---

**Agent reasoning**

```
Let me think about the correct approach. I need to manually expand
the squares and use the non-commutative ring structure.

The key lemmas I need are:
- `sq` : `sq x = x * x` or `pow_two` : `x^2 = x * x`
- `add_mul` : `(a + b) * c = a * c + b * c`
- `mul_add` : `a * (b + c) = a * b + a * c`
- `sub_mul` : `(a − b) * c = a * c − b * c`
- `mul_sub` : `a * (b − c) = a * b − a * c`

Let me construct the proof step by step. I'll need to be careful with
the algebra.

For `(a + b)^2`:
- `(a + b)^2 = (a + b) * (a + b)` (by `sq` or `pow_two`)
- `= a * (a + b) + b * (a + b)` (by `add_mul`)
- `= a * a + a * b + b * a + b * b` (by `mul_add` twice)
- `= a^2 + a * b + b * a + b^2` (by `pow_two`)

Now I need to show `a^2 + a * b + b * a + b^2 = a^2 + b^2`.

Since `a * b + b * a = 0`, we have `a^2 + a * b + b * a + b^2
= a^2 + 0 + b^2 = a^2 + b^2`.

Let me write this out carefully:

```lean
```

```
theorem pow_add_pow_eq {R : Type*} (a b : R) [Ring R]
    (h : a * b = - (b * a)) :
    (a + b) ^ 2 = a ^ 2 + b ^ 2 ∧ (a - b) ^ 2 = a ^ 2 + b ^ 2 := by
  have hab : a * b + b * a = 0 := by
    rw [h]
    simp [neg_add_cancel]
  constructor
  · simp only [sq, add_mul, mul_add]
    -- Now we have a * a + a * b + (b * a + b * b) = a * a + b * b
    -- Need to rearrange and use hab to cancel the middle terms
```
```

For the first part, I'll expand `(a + b)^2` using distributivity,
which gives me `a^2 + a*b + b*a + b^2`. The hypothesis `h` tells me
that `a*b = -(b*a)`, so `a*b + b*a = 0`. This means the middle terms
cancel, leaving just `a^2 + b^2`. The second part with `(a - b)^2`
follows similarly since `(-b)*a = -(b*a)` and `a*(-b) = -(a*b)`, so
the cross terms still sum to zero. The key insight is that after
expanding `(a - b)^2`, we get `a * a - a * b - b * a + b * b`, which
equals `a * a + b * b` precisely when `a * b + b * a = 0`. Since `hab`
gives us exactly this condition, the negation `-(a * b + b * a) = 0`
follows immediately.

For the proof structure, I should use `simp only` with the relevant
lemmas like `sq`, `sub_mul`, and `mul_sub` to handle the expansion,
then apply `hab` to eliminate the cross terms. The challenge is
ensuring the terms align properly after simplification - I may need to
use `calc` or explicit rewrites to guide the simplifier through the
algebraic steps. - `b) * (a - b)` becomes `a * (a - b) - b * (a - b)`,
which expands to `(a * a - a * b) - (b * a - b * b)`. Rearranging the
subtraction gives `a * a - a * b - b * a + b * b`. To reach
`a * a + b * b`, I need to eliminate the middle terms, which means
showing `- a * b - b * a = 0`. This follows directly from `hab` since
`-(a * b + b * a) = 0`.

Now I'll construct the proof using the ring structure. The first case
applies distributivity and uses `hab` to cancel the cross terms. The
second case handles the difference similarly, expanding `(a - b) ^ 2`
and applying the same cancellation. a = 0 := by
    rw [h, neg_add_cancel]
  have hab' : b * a + a * b = 0 := by
    rw [add_comm, hab]
  constructor
  · calc (a + b) ^ 2 = (a + b) * (a + b) := sq (a + b)
      _ = a * (a + b) + b * (a + b) := add_mul a b (a + b)
      _ = a * a + a * b + (b * a + b * b) := by rw [mul_add, mul_add]
      _ = a * a + (a * b + b * a) + b * b := by ring_nf
      _ = a * a + 0 + b * b := by

I need to establish that `a * b + b * a = 0` from the hypothesis, then
use this to simplify the expansion of `(a + b) ^ 2`. The calculation
expands the square, distributes multiplication, and rearranges terms
to group `a * b + b * a` together so it can be replaced with zero.
a * a + a * b) + (b * a + b * b) by distributing the multiplication.
Then I can rearrange using associativity to group the middle terms:
a * a + (a * b + b * a) + b * b. Since a * b + b * a = 0 by the
hypothesis, this simplifies to a * a + b * b.

Let me try a more direct approach using the distributive and
associative properties without getting bogged down in the details. I
could also check if there's a `noncomm_ring` tactic available, or just
work through the algebra step by step with basic lemmas.

For the second part with $(a - b)^2$, I'll need to expand it similarly and use the same relationship between a * b and b * a.

Expanding $(a - b)^2$ gives a * a - a * b - b * a + b * b. Since a * b + b * a = 0, I have -a * b - b * a = 0, so this simplifies to a * a + b * b as well. The key is applying the subtraction rules correctly and leveraging the anticommutative property throughout.
`+ b * b`
`= a * a + ((-a * b) + (-b * a)) + b * b`
`= a * a + (-(a * b + b * a)) + b * b`
`= a * a + (-0) + b * b`
`= a * a + 0 + b * b`
`= a * a + b * b`

I need to establish that negating a sum equals the sum of negations, that negating zero gives zero, and then apply these to collapse the middle terms. The key insight is recognizing that `a * b + b * a = 0` under the given hypothesis, so its negation becomes zero. From there, adding zero leaves just `a * a + b * b`.

Let me construct the proof:

```lean
theorem pow_add_pow_eq {R : Type*} (a b : R) [Ring R]
    (h : a * b = - (b * a)) :
    (a + b) ^ 2 = a ^ 2 + b ^ 2 ∧ (a - b) ^ 2 = a ^ 2 + b ^ 2 := by
  have hab : a * b + b * a = 0 := by rw [h, neg_add_cancel]
  constructor
  · rw [sq (a + b), sq a, sq b]
    rw [add_mul, mul_add, mul_add]
    rw [← add_assoc, add_assoc (a * a), hab, add_zero]
  · rw [sq (a - b), sq a, sq b]
    rw [sub_mul, mul_sub, mul_sub]
    rw [sub_sub, add_sub_assoc, ← sub_add, sub_add_eq_add_sub,
        hab, sub_zero]
```

The second part of the conjunction follows similarly by expanding $(a - b)^2$ and using the constraint that a*b + b*a = 0 to simplify the cross terms away. eq_add_neg (b * a)]
    rw [sub_eq_add_neg]
    rw [neg_add, neg_neg]
    rw [add_assoc, ← add_assoc (-(a * b)), hab_neg, zero_add]

The approach is becoming clearer–converting subtractions to additions with negatives, then using the commutativity property to rearrange terms and cancel them out.

Now I need to establish the core relationship: if `a * b = -(b * a)`, then `a * b + b * a = 0`. From there, I can expand both `(a + b)^2` and `(a - b)^2` using ring operations, apply the multiplication distributivity rules, and leverage the zero sum to show both expressions equal `a^2 + b^2`.

For the first part, after expanding `(a + b)^2` and distributing, the cross terms `a * b + b * a` cancel to zero. For the second part, I convert subtraction to addition of negatives, then use the fact that `-(a * b) + -(b * a) = 0` to eliminate the cross terms similarly.
= (a * a - a * b) + (-(b * a - b * b))` (applying sub_eq_add_neg to outer subtraction)

Now I need to distribute the negation: `-(b * a - b * b) = -(b * a)`

```
- (-(b * b)) = -(b * a) + b * b`

So the full expression becomes: `(a * a - a * b) + (-(b * a) + b * b)`

Now applying `sub_eq_add_neg` to the inner subtraction in the first
part: `a * a - a * b = a * a + (-(a * b))`

Final form: `(a * a + (-(a * b))) + (-(b * a) + b * b)
= a * a + (-(a * b)) + (-(b * a)) + b * b`
= (a * a - a * b) + (-(b * a - b * b))

For the negation of the subtraction, I need `-(b * a - b * b)
= -(b * a) + b * b`. The `neg_sub` lemma gives `-(a - b) = b - a`,
so applying it here yields `b * b - b * a`. Using commutativity of
addition, this becomes `-(b * a) + b * b`, which is what I'm after.
Let me verify the exact form of `neg_sub` in Mathlib to make sure I'm
applying it correctly. `b * b - b * a = b * b + (-(b * a))`, so
`-(b * a - b * b) = b * b + (-(b * a))`. I can rearrange this using
commutativity to get `-(b * a) + b * b`.

Now I can work through the full expression:
`(a * a - a * b) - (b * a - b * b)` becomes
`(a * a - a * b) + (-(b * a - b * b))` by converting subtraction to
addition of the negation, then `(a * a - a * b) + (b * b - b * a)`
by applying the negation rule, and finally
`(a * a + (-(a * b))) + (b *`

b - b * a)` by expanding the subtraction. Continuing with more
expansions: `(a * a + (-(a * b))) + (b * b + (-(b * a)))`. Using
associativity and commutativity, I can rearrange the middle terms so
that `-(a * b) + (-(b * a))` groups together, which equals zero by the
negation property. This leaves me with `a * a + (0 + b * b)`, which
simplifies to `a * a + b * b`.

Now I'll formalize this as a Lean proof. The strategy is to establish
that `a * b + b * a = 0` from the given condition, then derive the
negation property `-(a * b) + -(b * a) = 0`. With these lemmas in place,
I can expand both `(a + b) ^ 2` and `(a - b) ^ 2` and show they both
reduce to `a ^ 2 + b ^ 2`.

For the first case, expanding `(a + b) ^ 2` gives
`a * a + a * b + b * a + b * b`, which simplifies to `a * a + b * b`
once the middle terms cancel. The second case requires more careful
manipulation of subtraction and negation, converting `(a - b) ^ 2`
into additive form and applying the same cancellation principle to
reach the same result. -(b * b))` becomes `-(b * a) + b * b`

So the LHS is now `(a * a + -(a * b)) + (-(b * a) + b * b)`. I need to
rearrange this to get `a * a + b * b`. Using associativity, I can
group the negative terms: `a * a + (-(a * b) + -(b * a) + b * b)`.
Since `hab_neg` tells me that `a * b + b * a = 0`, the negatives cancel
out, leaving `a * a + (0 + b * b)`. `-(a * b) + -(b * a)) + b * b)`

Step 3: Apply `hab_neg` to combine the middle terms:
`a * a + (0 + b * b)`

Step 4: Simplify with `zero_add`:
`a * a + b * b`

The rewrites needed are `add_assoc` twice to reorganize the grouped
terms, then `hab_neg` to cancel the negated products, and finally
`zero_add` to eliminate the zero. Let me verify by tracking the
structure: starting with `(a * a + -(a * b)) + (-(b * a) + b * b)`, I
```

need to shift the parentheses so the two negated terms sit adjacent,
apply the hypothesis that they sum to zero, then clean up. standard
form is `add_assoc : a + b + c = a + (b + c)`. The goal
`(a * a + −(a * b)) + (−(b * a) + b * b) = a * a + b * b` has the
structure `(a + b) + (c + d)`, which doesn't directly match. I could
try `add_add_add_comm` if available, or work through it step by step
with a `calc` proof instead of relying on a single rewrite sequence.

I'll experiment with applying `add_assoc` directly or using
`← add_assoc` to shift parentheses. Using
`conv_lhs => rw [add_assoc, add_assoc]` might help flatten the left
side, transforming `(a * a + −(a * b)) + (−(b * a) + b * b)` into a
form where the terms can be rearranged more easily.
`−(a * b) + (−(b * a) + b * b)` becomes `(−(a * b) + −(b * a)) + b * b`
after applying `add_assoc`. Since `(−(a * b) + −(b * a)) = 0` by
`hab_neg`, this simplifies to `0 + b * b`, which equals `b * b` by
`zero_add`. The full expression then reduces to `a * a + b * b`.

I should try `rw [add_assoc, add_assoc, hab_neg, zero_add]` to handle
the nested additions, though I'm uncertain whether the rewrites will
penetrate the outer addition correctly. Let me attempt the proof and
observe what happens.

From the hypothesis that `a * b = −(b * a)`, I can derive
`a * b + b * a = 0`. Then for the first part, expanding `(a + b)²`
gives `a² + a*b + b*a + b²`, which simplifies to `a² + b²` once I use
the fact that `a*b + b*a = 0`. For the second part with `(a − b)²`,
the expansion yields `a² − a*b − b*a + b²`, and converting subtraction
to addition with negatives, I get `a² + (−(a*b)) + (−(b*a)) + b²`, which
also reduces to `a² + b²` using the derived relationship. oc,
add_assoc, hab_neg, zero_add]
```

Let me reconsider the naming conventions in Lean 4 / Mathlib.
`neg_add` should represent `−(a + b) = −a + −b` or similar, and
`neg_neg` should be `− (−a) = a`. I'll try this approach and see what
works.

I could also explore using `NoncommRing` tactics or the `abel` tactic
for abelian groups, though I'm uncertain if it handles multiplicative
structure properly.

Let me attempt a solution:

```lean
theorem pow_add_pow_eq {R : Type*} (a b : R) [Ring R]
    (h : a * b = − (b * a)) :
    (a + b) ^ 2 = a ^ 2 + b ^ 2 ∧ (a − b) ^ 2 = a ^ 2 + b ^ 2 := by
  have hab : a * b + b * a = 0 := by rw [h, neg_add_cancel]
  have hab_neg : −(a * b) + −(b * a) = 0 := by
    rw [← neg_add, hab, neg_zero]
  constructor
  · rw [sq (a + b), sq a, sq b]
    rw [add_mul, mul_add, mul_add]
    rw [← add_assoc, add_assoc (a * a),
        hab, add_zero]
  · rw [sq (a − b), sq a, sq b]
    rw [sub_mul, mul_sub, mul_sub]
    simp only [sub_eq_add_neg]
    rw [neg_add, neg_neg]
    rw [add_assoc, add_assoc, hab_neg, zero_add]
```

The first case expands `(a + b)²` and uses the anticommutativity
condition to show `a*b + b*a = 0`, which simplifies the cross terms.
The second case handles `(a − b)²` similarly, converting subtraction
to addition of negatives and applying the same anticommutativity
property. (a + b) * (a + b) = (a * a + a * b) + (b * a + b * b)`,
which is `(A + B) + (C + D)` where A = a*a, B = a*b, C = b*a,
D = b*b.

To show this equals `A + D` given `B + C = 0`, I need to rearrange.
The `← add_assoc` tactic rewrites `(x + y) + z` to `x + (y + z)`, but
my expression `(A + B) + (C + D)` has the form `X + Y` where X and Y
are themselves sums. I should apply `← add_assoc` with `a * a` as the
first argument to restructure the nested additions. a + b*b)`)
3. Now I need `a * b + (b * a + b * b) = b * b` given
   `a * b + b * a = 0`.
4. Using `← add_assoc` on `a * b + (b * a + b * b)`:
   `(a * b + b * a) + b * b`
5. Using `hab`: `0 + b * b`
6. Using `zero_add`: `b * b`

So the full LHS becomes `a * a + b * b` = RHS.

So for the first case, the rewrite sequence should be:
```lean
rw [add_assoc, ← add_assoc (a * b), hab, zero_add]
```

Let me redo the first case:
```lean
· rw [sq (a + b), sq a, sq b]
  rw [add_mul, mul_add, mul_add]
  -- Goal: (a * a + a * b) + (b * a + b * b) = a * a + b * b
  rw [add_assoc, ← add_assoc (a * b), hab, zero_add]
```

For the second case, I'll expand `(a − b)²` using the same approach.
After applying `sub_mul` and `mul_sub`, I get `(a * a − a * b)
− (b * a − b * b)`. Converting subtractions to negations and
simplifying with `neg_add` and `neg_neg` gives `(a * a + −(a * b))
+ (−(b * a) + b * b)`. Then I can rearrange using associativity and
apply `hab` to eliminate the cross terms, leaving just `a * a + b * b`.

```lean
· rw [sq (a − b), sq a, sq b]
  rw [sub_mul, mul_sub, mul_sub]
  simp only [sub_eq_add_neg]
  rw [neg_add, neg_neg]
  rw [add_ass
```

oc, ← add_assoc (−(a * b)), hab_neg, zero_add]
```

Now I can complete the proof by combining both cases. The first part
handles `(a + b) ^ 2` using the commutativity condition to simplify
the cross terms to zero. The second part applies the same strategy
to `(a − b) ^ 2`, using associativity and the derived lemma about
negated products to reach the same conclusion.
```lean
theorem pow_add_pow_eq {R : Type*} (a b : R) [Ring R]
    (h : a * b = − (b * a)) :
```

```
      (a + b) ^ 2 = a ^ 2 + b ^ 2 ∧ (a - b) ^ 2 = a ^ 2 + b ^ 2 := by
   have hab : a * b + b * a = 0 := by rw [h, neg_add_cancel]
   have hab_neg : -(a * b) + -(b * a) = 0 := by
     rw [← neg_add, hab, neg_zero]
   constructor
   · rw [sq (a + b), sq a, sq b]
     rw [add_mul, mul_add, mul_add]
     rw [add_assoc, ← add_assoc (a * b), hab, zero_add]
   · rw [sq (a - b), sq a, sq b]
     rw [sub_mul, mul_sub, mul_sub]
     simp only [sub_eq_add_neg]
     rw [neg_add, neg_neg]
     rw [add_assoc, ← add_assoc (-(a * b)), hab_neg, zero_add]
 ```
```

## Proposed proof

```
theorem pow_add_pow_eq {R : Type*} (a b : R) [Ring R]
    (h : a * b = - (b * a)) :
    (a + b) ^ 2 = a ^ 2 + b ^ 2 ∧ (a - b) ^ 2 = a ^ 2 + b ^ 2 := by
   have hab : a * b + b * a = 0 := by rw [h, neg_add_cancel]
   have hab_neg : -(a * b) + -(b * a) = 0 := by
     rw [← neg_add, hab, neg_zero]
   constructor
   · rw [sq (a + b), sq a, sq b]
     rw [add_mul, mul_add, mul_add]
     rw [add_assoc, ← add_assoc (a * b), hab, zero_add]
   · rw [sq (a - b), sq a, sq b]
     rw [sub_mul, mul_sub, mul_sub]
     simp only [sub_eq_add_neg]
     rw [neg_add, neg_neg]
     rw [add_assoc, ← add_assoc (-(a * b)), hab_neg, zero_add]
```

The proof goes again into the reviewer. This time, the code actually compiles and it is sent to the reviewer agent for a second check.

## Reviewer

### Compilation output

```
build_success
```

### User Input

```
<original>
```lean
theorem pow_add_pow_eq {R : Type*} (a b : R) [Ring R]
    (h : a * b = - (b * a)) :
    (a + b) ^ 2 = a ^ 2 + b ^ 2 ∧ (a - b) ^ 2 = a ^ 2 + b ^ 2 := by
      sorry
```
</original>

<proposed>
-- NEW PROOF
```lean
```

```
theorem pow_add_pow_eq {R : Type*} (a b : R) [Ring R]
    (h : a * b = - (b * a)) :
    (a + b) ^ 2 = a ^ 2 + b ^ 2 ∧ (a - b) ^ 2 = a ^ 2 + b ^ 2 := by
  have hab : a * b + b * a = 0 := by rw [h, neg_add_cancel]
  have hab_neg : -(a * b) + -(b * a) = 0 := by
    rw [← neg_add, hab, neg_zero]
  constructor
  · rw [sq (a + b), sq a, sq b]
    rw [add_mul, mul_add, mul_add]
    rw [add_assoc, ← add_assoc (a * b), hab, zero_add]
  · rw [sq (a - b), sq a, sq b]
    rw [sub_mul, mul_sub, mul_sub]
    simp only [sub_eq_add_neg]
    rw [neg_add, neg_neg]
    rw [add_assoc, ← add_assoc (-(a * b)), hab_neg, zero_add]
```
</proposed>
```

---

**Agent reasoning**

```
Let me analyze this carefully.

**Check 1: Statement Preserved?**

ORIGINAL statement:
```lean
theorem pow_add_pow_eq {R : Type*} (a b : R) [Ring R]
    (h : a * b = - (b * a)) :
    (a + b) ^ 2 = a ^ 2 + b ^ 2 ∧ (a - b) ^ 2 = a ^ 2 + b ^ 2
```

NEW PROOF statement:
```lean
theorem pow_add_pow_eq {R : Type*} (a b : R) [Ring R]
    (h : a * b = - (b * a)) :
    (a + b) ^ 2 = a ^ 2 + b ^ 2 ∧ (a - b) ^ 2 = a ^ 2 + b ^ 2
```

These are identical. Check 1: True

**Check 2: No Sorry in NEW PROOF Body?**

Looking at the proposed proof body:
```lean
  have hab : a * b + b * a = 0 := by rw [h, neg_add_cancel]
  have hab_neg : -(a * b) + -(b * a) = 0 := by
    rw [← neg_add, hab, neg_zero]
  constructor
  · rw [sq (a + b), sq a, sq b]
    rw [add_mul, mul_add, mul_add]
    rw [add_assoc, ← add_assoc (a * b), hab, zero_add]
  · rw [sq (a - b), sq a, sq b]
    rw [sub_mul, mul_sub, mul_sub]
    simp only [sub_eq_add_neg]
    rw [neg_add, neg_neg]
    rw [add_assoc, ← add_assoc (-(a * b)), hab_neg, zero_add]
```

No ``sorry'' or ``admit'' found. Check 2: True

**Check 3: No Other Issues?**
```

```
Examining the proof structure: the first `have` establishes that
`a * b + b * a = 0` using the anticommutativity hypothesis, then
the second `have` derives a related fact about negated terms. The
constructor splits the conjunction into two goals, each handled by
expanding the square definitions and applying ring operations to
simplify using the established equalities. `sub_eq_add_neg` converts
subtraction to addition with negation, `neg_neg` shows double negation
cancels, and `zero_add` is the left identity for addition. These are
all standard Mathlib lemmas. The proof strategy is straightforward: it
derives `a * b + b * a = 0` from the hypothesis, then establishes
`-(a * b) + -(b * a) = 0`, uses the constructor to handle the
conjunction, and simplifies each component through algebraic
manipulation. No undefined references or logical gaps appear present.
```

---

**Review result**

```
check_1: true
check_2: true
check_3: true
approved: true
```

---

Given the successful review, the proof is completed and validated.

## D. Ablation dataset

In Table 3, we report the list of 100 problems randomly extracted from the Putnam competition to run our ablation studies.

## E. Error analysis

We analyze the traces of the different ablation studies conducted in Sections 4.1 and 4.2 to understand the differences between approaches by quantifying their errors. To do so, we take every failed proof attempt and extract the first error in the proposed code, provided that some types of errors can cause multiple subsequent related ones like a syntax error. Then, we assign this error to one of the following six categories based on the build output message:

- **Bad import**: The file tries to load a module that Lean cannot find.

- **Name error**: The code refers to an identifier that does not exist in scope.

- **Sorry remains**: The proof contains a `sorry` or `admit` placeholder.

- **Syntax error**: The code contains invalid Lean syntax.

- **Tactic failure**: The syntax and names are correct but a tactic cannot do what it is asked.

- **Unsolved goals**: The proof has open goals.

These error categories also help gauge the quality of the generated code, since they reflect the depth at which the verification failed. In general, lower-level errors such as bad imports, syntax errors, or unknown names typically prevent Lean from reaching proof checking, whereas tactic failures and unsolved goals usually arise only after the surrounding declaration has parsed and elaborated successfully.

### E.1. Ablation comparison

In Figure 4, we show the quantitative error analysis for the different ablations considered in Section 4.1. The evaluated system uses Claude Opus 4.5 with a thinking budget of 10k tokens and a maximum of 100 iterations per problem. We

*Table 3.* 100 Randomly-Selected PutnamBench Samples for Ablation Studies (Sorted)

| | | | |
|---|---|---|---|
| putnam_1962_a1 | putnam_1962_b6 | putnam_1963_a3 | putnam_1964_a4 |
| putnam_1965_b4 | putnam_1966_a6 | putnam_1966_b6 | putnam_1967_a1 |
| putnam_1968_b5 | putnam_1970_b3 | putnam_1970_b6 | putnam_1972_b4 |
| putnam_1972_b6 | putnam_1973_a3 | putnam_1973_a6 | putnam_1974_a1 |
| putnam_1974_a6 | putnam_1974_b3 | putnam_1974_b6 | putnam_1975_b5 |
| putnam_1976_b3 | putnam_1978_b4 | putnam_1980_a3 | putnam_1980_a6 |
| putnam_1980_b5 | putnam_1981_b1 | putnam_1982_a2 | putnam_1982_b2 |
| putnam_1982_b3 | putnam_1982_b5 | putnam_1984_a2 | putnam_1985_a6 |
| putnam_1986_a1 | putnam_1986_a3 | putnam_1986_a5 | putnam_1986_b3 |
| putnam_1987_a6 | putnam_1987_b1 | putnam_1988_b3 | putnam_1988_b6 |
| putnam_1989_a1 | putnam_1989_a2 | putnam_1990_a4 | putnam_1990_a6 |
| putnam_1991_a4 | putnam_1991_a5 | putnam_1991_b1 | putnam_1991_b2 |
| putnam_1992_a2 | putnam_1992_b3 | putnam_1994_a6 | putnam_1994_b3 |
| putnam_1995_b1 | putnam_1995_b6 | putnam_1996_a4 | putnam_1996_b3 |
| putnam_1996_b4 | putnam_1997_a3 | putnam_1997_b2 | putnam_1998_a4 |
| putnam_1999_a3 | putnam_2000_a2 | putnam_2000_a5 | putnam_2000_b1 |
| putnam_2001_a1 | putnam_2003_a5 | putnam_2003_b1 | putnam_2004_a1 |
| putnam_2004_b4 | putnam_2005_a3 | putnam_2005_a5 | putnam_2007_a5 |
| putnam_2008_b4 | putnam_2009_b1 | putnam_2010_a2 | putnam_2010_a3 |
| putnam_2010_a4 | putnam_2012_a2 | putnam_2012_a5 | putnam_2012_a6 |
| putnam_2012_b5 | putnam_2014_a4 | putnam_2014_a6 | putnam_2014_b2 |
| putnam_2014_b3 | putnam_2015_a6 | putnam_2015_b6 | putnam_2016_b4 |
| putnam_2018_a3 | putnam_2018_a5 | putnam_2018_b5 | putnam_2019_b4 |
| putnam_2020_a6 | putnam_2020_b4 | putnam_2020_b6 | putnam_2021_a2 |
| putnam_2021_b2 | putnam_2022_b3 | putnam_2023_a6 | putnam_2024_a3 |

observe that this model does not fail to write the correct imports, unlike other models that we discuss in the following Appendix E.2.

The various improvements to the agentic system (context management and search tools) increase the success rate for the same number of iterations reducing the amount of failed proofs. However, not all errors are mitigated equally, which is reflected in a shift on the error distribution. In particular, we observe it moves away from low-level errors such as name errors, dropping from 13% to 7%, to higher level ones like unsolved goals, increasing from 27% to 36%. This is because the total number of low-level errors is drastically reduced (name errors drop by 53% and syntax errors by 26% between feedback and full system), but the total number of higher-level errors remains the same (+3% unsolved goals), suggesting that most of the progress comes from addressing unsolved goals. Indeed, the performance of each of the tested systems is correlated with the relative amount of unsolved goal errors: the larger the fraction of errors that fall into this category, the higher the model performance. As we explain above, these errors can only appear when the code is in a good state overall, and they provide the most informative feedback for the development of the proof in subsequent iterations.

This also allows us to obtain a better understanding of the effect of the different memory mechanisms. In both cases, we observe a shift towards higher-level errors. However, the self-managed context based on self-reflection helps to reduce the amount of syntax errors and tactic misuses, which is translated into a larger fraction of unsolved goal errors than with the history of the most recent attempts. We see this effect even more pronounced when incorporating search tools on top of the self-reflection memory mechanism, which reduce the overall time the agent spends grappling with Lean (naming, syntax, proper tactics usage) and increases the fraction of the time it invests on developing the proof by addressing the open goals.

An alternative strategy that the agent has to explore the state of different goals across the proof is by placing a `sorry` in the target locations, as we explain in Section 3. We can monitor this behavior through the number of errors related to remaining sorries. The usage is rather homogeneous overall across systems, with the history mechanism using it the least in relative terms.

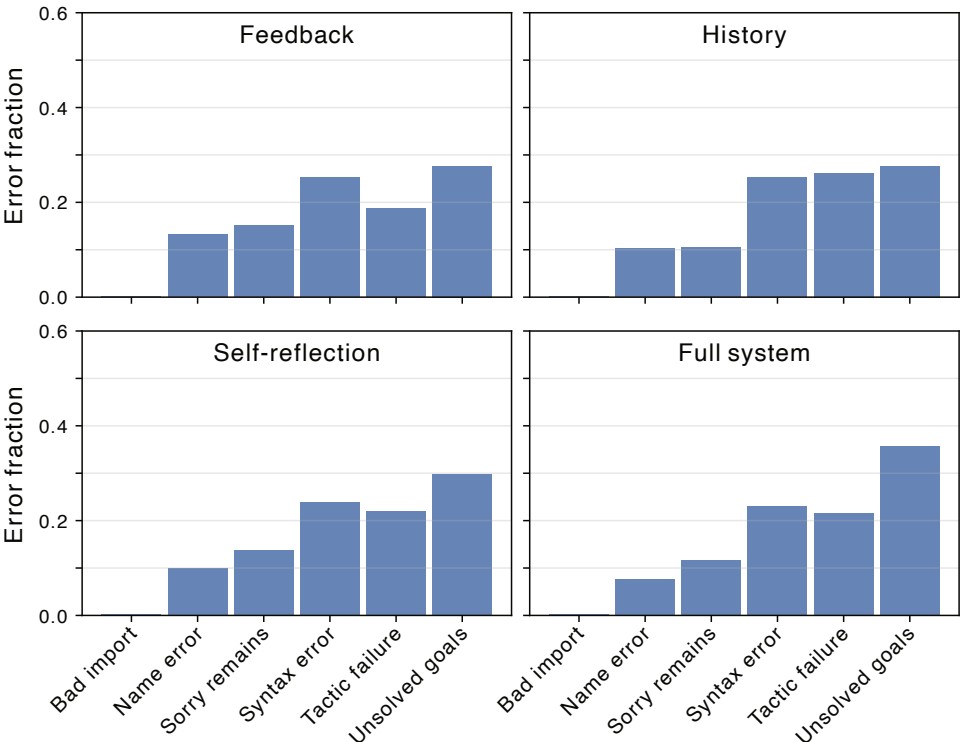

*Figure 4.* **Error analysis across multiple ablations**. Error distribution for each of the system ablations studied in Section 4.1: agent with feedback, history of the previous 5 proof attempts, self-managed context based on self-reflection, and the full system with self-reflection and search tools.

### E.2. Foundation model comparison

In Section 4.2, we observe clear differences in the performance of AxProverBase with different underlying LLMs. We show the error distribution for each of the considered models in Figure 5, where we can observe a clear difference between the Gemini 3 and the Claude 4.5 model families, and the same shift towards higher-level errors discussed in Appendix E.1 between the weaker and stronger model within the same family.

Perhaps the most evident difference between models is the number of import errors that the Gemini 3 models incur, compared to the near complete absence of those in the Claude 4.5 family. We observe that 56% and 33% of the failed proof attempts of Gemini Flash and Pro, respectively, contain bad imports as opposed to 1% and 0.4% for Claude Sonnet and Opus. We can directly compare these numbers since the statistics are from the first error of each failed proof attempt and imports are always at the top of the file.

Between the same model families, we observe a shift in the error distribution from trivial to deeper ones for the frontier models. This is due to stronger foundation models making significantly less import, naming and syntax errors in absolute terms, while making the same number or even more tactic failures and unsolved goals over the same tasks. This is the same effect that improvements in the agentic architecture (described in Appendix E.1) had in the agent's performance: the agent spends fewer iterations struggling with Lean, thus increasing the resources spent in developing the proof. The composition of both effects, better model and architecture, explains the observation in Figure 3 that more advanced models benefit more from the proposed agentic framework.

## F. Cost-performance analysis

We study the cost-performance relationship of the configuration of AxProverBase used to conduct the evaluations in Section 4.3. We consider the success rate as performance metric, and we take the number of tokens as a measure of the cost that allows us to compare different methods. Hence, we provide the Pareto curve of success rate vs the maximum number of tokens allowed per problem for every dataset. We break down the total number of tokens into input (prompt)

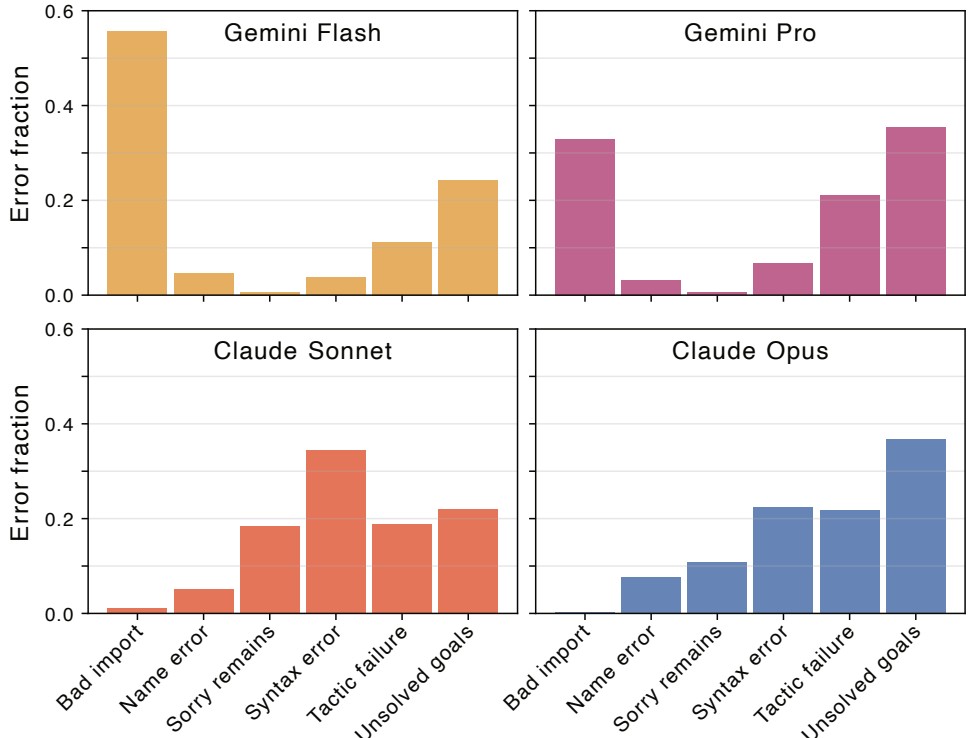

*Figure 5.* **Error analysis across multiple underlying LLMs**. Error distribution for each of the underlying LLMs studied in Section 4.2: Gemini 3 Flash, Gemini 3 Pro, Claude 4.5 Sonnet and Claude 4.5 Opus.

| Dataset | Success rate | Max tokens | Avg tokens | Output token % (STD) |
|---------|-------------|------------|------------|----------------------|
| PutnamBench | 54.7% | 4.2M | 1.4M | 29.7 (6.2) |
| LeanCat | 59.0% | 1.9M | 0.6M | 26.2 (8.1) |
| FATE-M | 98.0% | 1.4M | 55.4k | 28.6 (11.7) |
| FATE-H | 66.0% | 2.8M | 1.0M | 28.5 (4.5) |
| FATE-X | 24.0% | 2.9M | 1.3M | 28.7 (3.7) |

*Table 4.* Cost statistics for the performance reported in Table 1. We provide the success rate for the dataset, together with the maximum tokens per problem with which it is obtained before saturating performance, the average tokens per problem within the budget including failed proofs, and the mean ratio between input and output tokens (and its standard deviation across problems).

and output (generated) tokens, provided that they typically entail significantly different costs. Additionally, we compute the average tokens per problem, and include data from other existing approaches referenced through the main text when available. Table 4 contains a summary of the statistics for each dataset.

### F.1. PutnamBench

In Figure 6, we show the cost-performance analysis for the evaluation over the PutnamBench dataset (Tsoukalas et al., 2024). The 54.7% reported success rate in Table 1 is achieved with 4.2M maximum tokes per problem, with an average of 1.4M tokens per problem. The average fraction of output tokens is 29.7% (6.2 STD), meaning that there are around 2.4 times more input than output tokens, on average.

We compare these metrics with those reported by Hilbert (Varambally et al., 2026), which achieves a pass@1 success rate of 55.9% with a maximum of 1880.4M tokens per problem. The Pareto frontier for both methods shows a significantly better scaling of AxProverBase, which operates at a fraction of the cost for the same performance. For instance, Hilbert matches the maximum performance achieved by AxProverBased with a maximum token budget that is about two orders of magnitude larger. Similarly, for a fixed budget of 1M tokens per problem, AxProverBase achieves a ~40% success rate compared to Hilbert's ~15%.

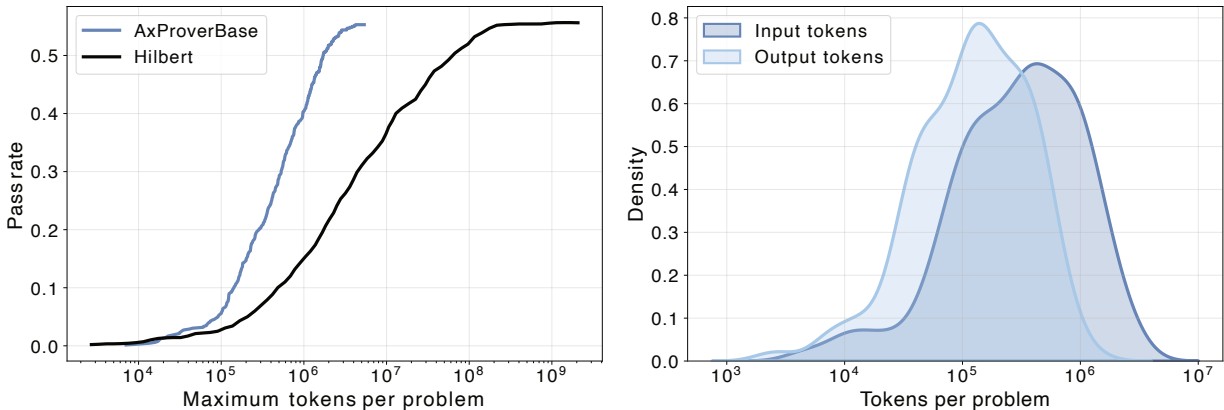

*Figure 6.* **Cost-performance analysis for the PutnamBench dataset evaluation.** On the left, we show the cost-performance Pareto frontier in terms of proof pass rate vs maximum tokens per problem for AxProverBase and Hilbert prover (Varambally et al., 2026). AxProverBase shows a better scaling than Hilbert, using two orders of magnitude fewer tokens at their highest performance. On the right, we show the distribution of input and output tokens with an average input/output ratio of 2.4.

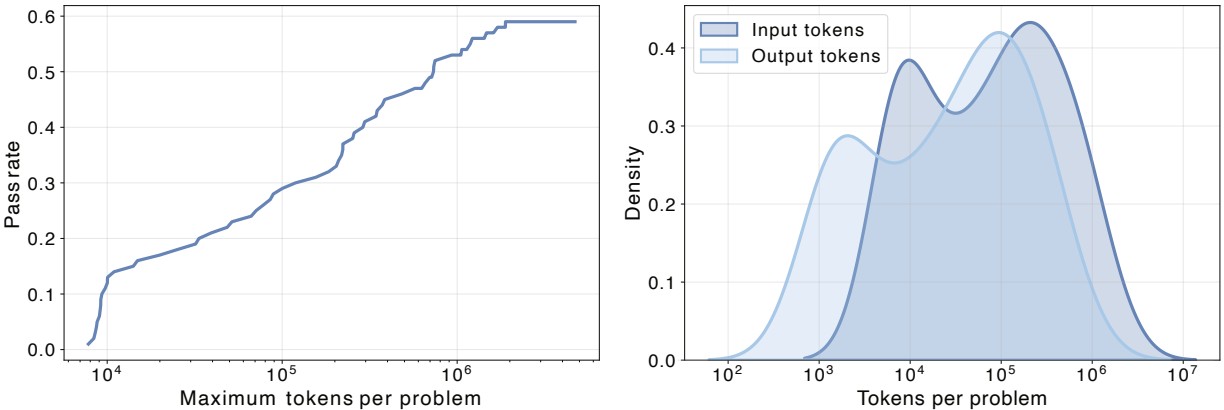

*Figure 7.* **Cost-performance analysis for the LeanCat dataset evaluation.** On the left, we show the cost-performance Pareto frontier in terms of proof pass rate vs maximum tokens per problem. On the right, we show the distribution of input and output tokens, with an average input/output ratio of 2.7.

## F.2. LeanCat

In Figure 7, we show the cost-performance analysis for the evaluation over the LeanCat dataset (Xu et al., 2025). The 59.0% success rate reported in Table 1 is achieved with 1.9M maximum tokens per problem, with an average number of tokens per problem of 0.6M. On average, the output tokens represent a 26.2% (8.1 STD) of the total tokens, slightly less than in the PutnamBench and FATE datasets.

## F.3. FATE

In Figure 8, we show the cost-performance analysis for the evaluation over the full FATE dataset (Jiang et al., 2026). The reported 98.0%, 66.0% and 24.0% success rates respectively for the FATE-M, FATE-H and FATE-X datasets in Table 1 are obtained with 1.4M, 2.8M and 2.9M maximum tokens per problem. The average number of tokens per problem is 55.4k, 1.0M and 1.3M, respectively. Their average (STD) output token fractions are 28.6% (11.7), 28.5% (4.5) and 28.7% (3.7).

We compare these metrics with those provided by FATE (Jiang et al., 2026) in their evaluation of multiple automated theorem provers based on: DeepSeek-R1 (Guo et al., 2025), Qwen3 (Yang et al., 2025), DeepSeek-Prover-V2 (Ren et al., 2025), Goedel-Prover-V2 (Lin et al., 2026), and Kimina-Prover (Wang et al., 2025a). The FATE report provides the performance of these methods under a normalized computational budget accounting for model size, average output tokens, and the number of passes per problem. The input tokens are discarded as they are significantly shorter than the output. Therefore, we

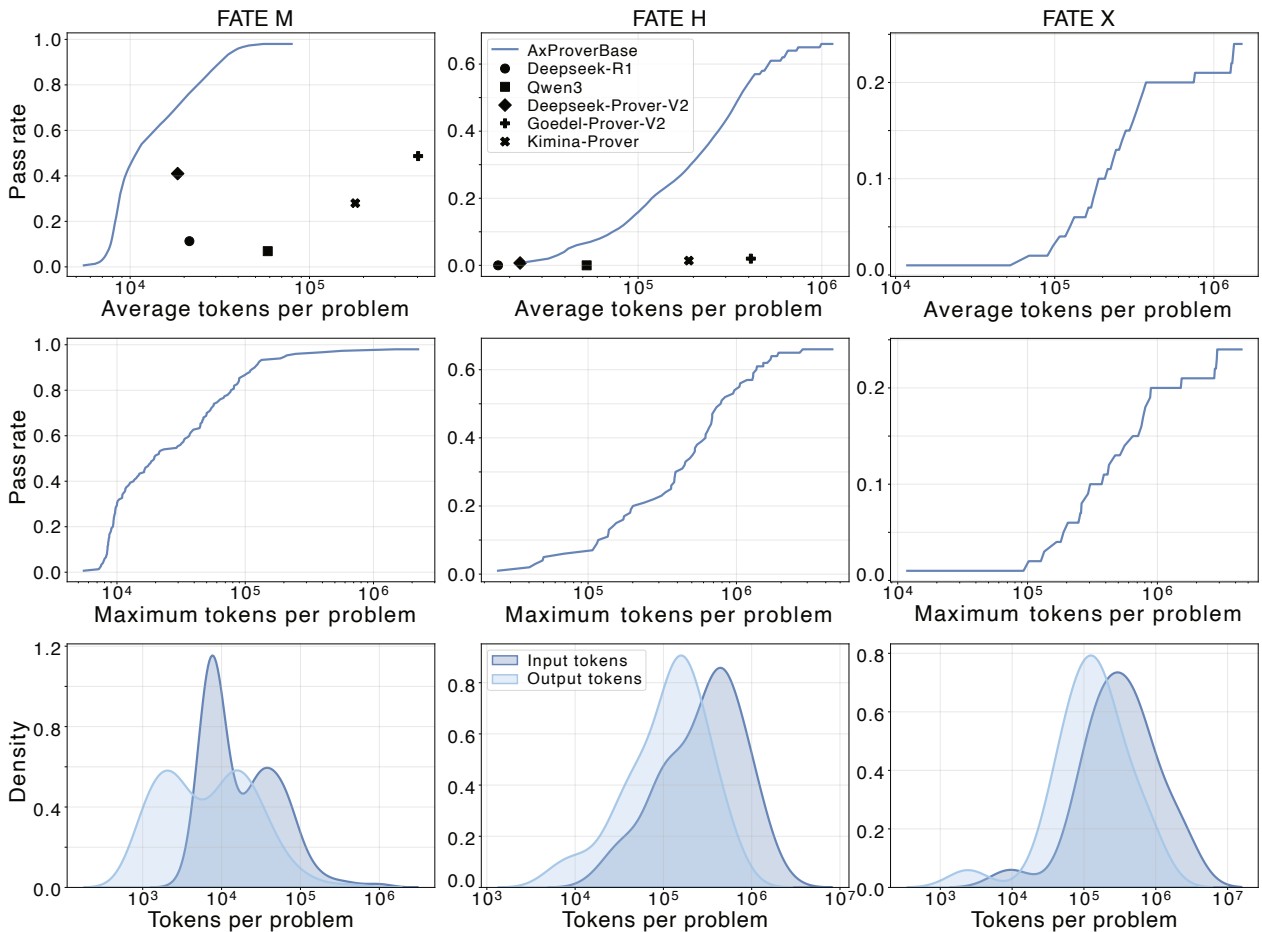

*Figure 8.* **Cost-performance analysis for the FATE dataset evaluation.** Each column corresponds to one of the splits of FATE: M, H, and X from left to right. For each of those, we show (top to bottom): the pass rate vs average tokens per problem, the Pareto frontier of pass rate vs maximum tokens per problem, and the input-output token distributions.

compute the average tokens per problem for the given success rate as the average response length times the number of passes for each model. We report these metrics in Figure 8 for the FATE-M and H datasets, provided that none of these models achieved a successful proof.

Among these methods, the specialized theorem provers (DeepSeek, Goedel and Kimina provers) outperform the general-purpose ones (Qwen3 and DeepSeek-R1), which cannot prove any problem in the FATE-H dataset. Perhaps the most illustrative case is between the two DeepSeek models: where Prover-V2 achieves 4x the success rate of R1, having the same size and using a similar number of tokens per problem. However, we observe that AxProverBase vastly outperforms all of these methods for the same number of average tokens per problem, even when only a small fraction of those are output tokens. The differences become especially evident in the harder datasets where these methods can barely prove any problem.

