# OpenReview forum: "A Minimal Agent for Automated Theorem Proving"
_ICML.cc/2026/Conference — ICML 2026 regular_

### Official Review · Reviewer_1afg · 2026-02-21

**Soundness:** 3
**Presentation:** 3
**Significance:** 2
**Originality:** 2
**Overall Recommendation:** 5
**Confidence:** 3

**Summary:**

The paper presents a prompting based workflow for automated theorem proving in Lean. The paper designs a modular workflow consisting of a proposer agent which generates proofs, an LLM based review system, a memory layer. Then, it studies the best design decisions for each component showing that iterative refinement is better than single shot generation, and different frontier models perform drastically differently. The analysis shows that using this agentic workflow leads to state of the art performance on several theorem proving benchmarks.

**Compliance With Llm Reviewing Policy:**

Affirmed.

**Final Justification:**

After reading author's rebuttal and other reviews, I decide to maintain positive recommendation to the paper.

**Key Questions For Authors:**

1. Is the review system implemented as an LLM? It feels like to prevent cheating, the best way is to have a more comprehensive rule-based system since LLM judge could be subject to prompt injection.

2. What does the axis of Figure 3 (b) mean exactly? Is the cost normalized to each model maximum cost or the overall max cost?

**Limitations:**

yes

**Strengths And Weaknesses:**

The paper is technically sound with sufficient experimental results. I believe a simple enough baseline for theorem proving is necessary for future research.

The paper is also generally well-written and I have no trouble understanding the paper. Only a few design decisions are unclear.

Because the paper is simple to implement and does not assume finetuning access of models, it should also be widely applicable.

---

> ### Author Rebuttal · Authors · 2026-03-31
>
> Thank you for your kind comments and revision. We are happy to see that you appreciate the strengths of our contribution, and we will address your questions one by one:
>
> We are indeed implementing a rule-based system that covers most of the cases. We use the Lean compiler and tools like [LeanInteract](https://augustepoiroux.github.io/LeanInteract/stable/) to deterministically check the resulting Lean code. Only after passing this first inspection, we use an LLM as an additional safety measure to capture any other cheating mechanisms that we may have overlooked or more advanced cheating schemes like using metaprogramming tricks, although these are generally uncommon because they require the proposer agent to actively try to cheat. For example, in our early development, we had not accounted for the possibility that Mathlib’s search tactics like `apply?`, `exact?` or `rw?` may produce a proof that successfully compiles while being incomplete, and the LLM layer caught this. This allowed us to include checks for these kinds of cases in the first review layer, so they do not get to the LLM anymore. We believe that this approach of having the code first pass a set of strict deterministic checks before being reviewed by an LLM is quite robust to prompt injection given that the code that reaches the LLM has already been scrutinized.
>
> The cost in Figure 3 (b) refers to the actual money spent running each of the models including both proven and unproven problems. We normalize the costs with respect to the most expensive experiment (Claude Opus 4.5 with up to 100 iterations per problem and 10k thinking tokens) to perform a relative comparison between them, especially between models of the same family. We make this clear in the updated version of the paper. We decided to go for dollar cost because we were comparing models with very different capacities that also produced different quantities of input and output tokens (for instance with more or less reasoning budgets) and the cost serves as a quantity that encompasses all of these aspects. This allowed us to identify the most cost-effective strategy to run our final evaluation: having two settings that yield the same maximum performance with Claude Opus 4.5 100 iterations and 10k thinking budget or 50 iterations and 32k thinking budget, the latter is nearly 20% cheaper.
>
> We are also happy to share that our reviewed version of the manuscript includes:
> - A direct comparison of token cost between our approach and state of the art methods under equal conditions. For example, we show that we achieve the same pass@1 performance as Hilbert on PutnamBench using about 200 times less tokens.
> - A cost-performance analysis of our approach for all datasets including the pareto frontier of success rate vs tokens as well as a breakdown of input-output tokens for the model.
> - A quantitative analysis of the failure modes. For example, we show that more than half of the errors of Gemini Flash and about a third of the Gemini Pro contain hallucinated imports, whereas Claude models hardly ever write bad imports (1.4% and 0.4% of errors for Sonnet and Opus respectively).
> - Various improvements in the formatting and discussion sections.
>
> Overall, we believe that the paper has significantly improved in the quality of its exposition as well as in the significance of its results with the new analysis (see the second paragraph in the rebuttal of Reviewer 1 for more details).

---

> > ### Author Rebuttal · Reviewer_1afg · 2026-04-02
> >
> > I decide to keep my positive rating.

---

### Official Review · Reviewer_BzvB · 2026-03-09

**Soundness:** 2
**Presentation:** 2
**Significance:** 3
**Originality:** 2
**Overall Recommendation:** 2
**Confidence:** 3

**Summary:**

This paper proposes AxProverBase, a minimal agentic framework for automated theorem proving in Lean 4. The authors argue that the dominant sources of performance in current AI-based provers can be distilled into three components: iterative proof refinement driven by compiler feedback, a memory mechanism to prevent cyclic failures across iterations, and access to library/web search tools. Rather than proposing a new algorithmic innovation, the paper's contribution is empirical: it conducts a bottom-up ablation study to quantify the individual impact of each component, compares several frontier LLMs within this framework, and evaluates the resulting system on PutnamBench, FATE, and LeanCat benchmarks. The authors claim that this simple architecture achieves performance competitive with significantly more complex SOTA systems.

**Compliance With Llm Reviewing Policy:**

Affirmed.

**Final Justification:**

Thank the authors for the detailed rebuttal and clarifications. The response helped resolve several presentation-level issues and made the paper’s intentions clearer.

That said, I remain concerned about the overall maturity of the submitted manuscript. In my view, the current version still feels underdeveloped as a submission: some important empirical support was added in rebuttal or deferred to the camera-ready version, and several issues in presentation and framing go beyond what I would consider a normal camera-ready pass. I do not think these are merely minor polishing matters.

For that reason, while I appreciate the value of the direction and the authors’ effort in addressing questions, I keep my original score unchanged.

**Key Questions For Authors:**

See Weaknesses.

**Limitations:**

The authors acknowledge several limitations in Section 5, including opportunities to improve individual components (memory, reviewer, library search) and the potential benefit of fine-tuned specialized models.

**Strengths And Weaknesses:**

### Strengths

- S1. **Timely and practically motivated contribution.** The observation that recent general-purpose LLMs have closed much of the gap with specialized fine-tuned provers is well-motivated. A clean baseline that isolates the contribution of scaffolding components is genuinely useful for the community, particularly given how rapid Lean and Mathlib evolve and how quickly fine-tuned models become outdated.

- S2. **Ablation study covers the right questions.** The evaluation covers several meaningful design questions for the field: how much does iterative refinement matter over single-shot sampling? How do different memory strategies compare? How much do search tools add on top? The general direction of the findings (refinement > memory > tools) is plausible and informative.

- S3. **Cost-awareness.** The paper's explicit attention to cost-performance tradeoffs (average $12.6/sample, comparison of thinking budgets) is a practical contribution that is often neglected in prior theorem-proving literature.

### Weaknesses

- W1. **Critical internal inconsistency in Figure 2.** The paper states that ablation studies are conducted on 100 randomly selected PutnamBench samples, yet the caption of Figure 2 states "we use two samples for the different ablations, and three for the full system." These two statements are irreconcilable as written. If "two samples" refers to the number of parallel proof attempts per iteration rather than the number of problems, this distinction is never clarified in the text. This is a serious flaw in a paper whose primary contribution is empirical.

- W2. **Unfair and misleading comparison with Hilbert.** The paper repeatedly compares AxProverBase at 20 or 100 iterations against Hilbert at pass@1, claiming performance of "double," "triple," and "quadruple." However, Hilbert's full configuration operates at pass@1840, achieving 462/672 on PutnamBench (the former leaderboard top position). Comparing an iterative system at 100 attempts against a competitor's pass@1 is not a meaningful benchmark comparison. The actual computational budgets are not equalized, and the paper provides no apples-to-apples comparison with Hilbert's full configuration.

- W3. **"Cost" is never formally defined.** Figure 3(b) uses "Normalized cost relative to maximum" as its x-axis, but the paper never specifies what cost means: token count, wall-clock time, or monetary expenditure. The claim in Section 5 that "execution time has consistently been an order of magnitude lower than Hilbert" is stated without any supporting numbers, making it an unsupported assertion.

- W4. **Absence of quantitative failure analysis.** The paper identifies several important failure modes qualitatively (e.g., cyclic errors without memory, Gemini models hallucinating outdated Lean/Mathlib APIs), but provides no quantitative breakdown. What fraction of failures are due to API hallucination vs. proof strategy errors? At what iteration does cycling behavior typically onset? These observations are presented as findings but are backed by no data.

- W5. **Unclear relationship to Ax-Prover.** The PutnamBench leaderboard lists "Ax-Prover" (Breen et al., 2025) at rank 7, solving 91/672 problems at pass@1 with ~100 tool calls. This paper's system is named "AxProverBase," is similarly described as a multi-agent Lean prover, and cites the Ax-Prover paper. However, the paper never explicitly discusses the relationship between these two systems. If AxProverBase is a simplified version of Ax-Prover, the substantially higher reported performance (54.7% vs. 13.8% on PutnamBench pass@1) requires explicit explanation. If they are unrelated, the naming choice is confusing and the comparison should be made explicit.

- W6. **Minor: inconsistent terminology.** The paper alternates between "LeanCat" and "LeanCAT" to refer to the same dataset.

---

> ### Author Rebuttal · Authors · 2026-03-31
>
> Thank you for the positive assessment of the paper and the useful comments. We sincerely believe that our manuscript has improved as we have addressed the main weaknesses you identified incorporating new quantitative data analysis that respond to your concerns:
>
> **W1**.
> We select a random sample of 100 problems from PutnamBench, which is fixed across all our ablation studies and the concrete list of problems is specified in Appendix A for reproducibility. Subsequently, we run the agent twice for the different ablations and three times for the full system to get statistics on the performance over the fixed 100 problems. We clarify this in the camera ready version.
>
> **W2**.
> We apologize for the confusion regarding iterations vs. realizations. In our framework, iterations dictate the depth of the internal refinement loop for a single proof attempt (similar to Hilbert's recursion depth or the test-time scaling parameter of Seed-Prover). It is distinct from k in pass@k, which denotes independent parallel realizations. Hence, both AxProverBase and Hilbert propose multiple proofs within a single realization, and a realization is terminated when they either complete the proof successfully or exhaust their budget. Therefore, it is fair to compare them under similar conditions.
>
> A recently updated version of Hilbert’s paper (March 16th 2026) reports both pass@1 on PutnamBench, as well as pareto frontier curves of success rate vs maximum token budget per problem, which was not available at the time of writing our paper. With this new data, we have updated our analysis with an apples-to-apples comparison between AxProverBase and Hilbert on the full PutnamBench at matched cost. AxProverBase and Hilbert achieve a pass@1 success rate in PutnamBench of 54.7% and 55.9% at 5.4 and 1880.4 max million tokens per problem, respectively.
>
> We believe these new results significantly strengthen the main points of our paper. We have achieved a competitive performance at a fraction of the tokens by distilling the main elements contributing to performance with the help of ablations. This is clear evidence that the field would significantly benefit from strong baselines like AxProverBase that capitalize on the progress of frontier models, against which complex architectures should be fairly evaluated. You may find exemplary figures for these results [here](https://anonymous.4open.science/r/icml-axproverbase/comparison.md), together with an analogous analysis on the FATE dataset that includes cost analysis of several provers (also freshly released data).
>
> Besides this comparison, we provide an additional cost-performance analysis of our method for all the datasets in the updated version of the paper. We also separate input and output tokens given that they imply significantly different costs, and provide dollar cost per problem metrics.
>
> **W3**.
> The cost in Figure 3 (b) refers to the money spent running each of the models including both proven and unproven problems. We normalize the costs with respect to the most expensive experiment to perform a relative comparison between them, especially between models of the same family.
>
> We have replaced our previous qualitative execution time claim with the new comparison analysis using normalized cost metrics like maximum tokens per problem (see W2 above).
>
> **W4**.
> We have included a quantitative analysis of the failure modes for the different ablations. Particularly, we show that hallucinated imports are present in 56.1% and 32.4% of the failed proof attempts of Gemini Flash and Pro, respectively, while only in 1.4% and 0.4% for Claude Sonnet and Opus. You can find a summary table for these results and an exemplary figure of the additional failure mode breakdown that will be included in the revised version of the paper [here](https://anonymous.4open.science/r/icml-axproverbase/failures.md).
>
> Regarding the repetition of mistakes, we realize that using the word “cycle” may have been misleading. Quantifying these requires expert supervision, given that they are often related to abstract concepts (e.g. proof strategy) that do not have a direct identifier in the code or build output. Hence, we cannot provide a complete systematic analysis of these and we have accordingly rephrased the text.
>
> **W5**.
> Our work draws inspiration from Ax-Prover to use simple approaches (off-shelf model with tools), and to capitalize on the power of iterative self-correction. However, AxProverBase has a very different architecture and should not be considered a simplification of the other.
> Ax-Prover is a ReAct-style agent with tools that include Lean build and file write among several others. The agent can choose when to build and verify the code.
> On the other hand, AxProverBase is based on a predefined workflow, where a proof is always proposed and subsequently reviewed, to ensure it actually proves the theorem.
> We will make this clear in the paper.
>
> **W6**.
> Thank you for catching this. Fixed!

---

> > ### Author Rebuttal · Reviewer_BzvB · 2026-04-02
> >
> > Thank the authors for their explanation. The rebuttal partially addresses my concerns, but does not resolve them sufficiently to change my overall assessment. Some clarification is helpful, especially on presentation issues such as the Figure 2 inconsistency, the definition of cost, and the relationship to Ax-Prover. However, key empirical support appears to rely on substantial new analysis introduced during the rebuttal, rather than on evidence already present in the original submission. This weakens the extent to which the rebuttal can resolve concerns about the submission itself.
> >
> > More importantly, I remain unconvinced that the paper has adequately disentangled the contribution of the (minimal) agent framework from the contribution of the underlying frontier model. The strongest results and much of the “cost-effective near-SOTA” framing rely on Claude Opus 4.5, and I do not think the rebuttal sufficiently justifies how much of the reported performance should be attributed to the proposed framework versus the backbone model. I also note that a similar concern was raised elsewhere in the discussion, and I find that concern well-founded. The response argues that open-source comparisons such as Qwen3 are currently difficult to run reliably, but this explains the missing evidence rather than resolving the attribution issue.
> >
> > For these reasons, I mark the concerns as only partially resolved and keep my score unchanged.

---

> > > ### Author Response · Authors · 2026-04-02
> > >
> > > We thank you for the feedback. We address your main two points in the following two sections.
> > >
> > > # Previous concerns
> > > We would like to clarify that the analysis presented in the rebuttal was intended to address the expressed concerns in the original review and they are complementary data that do not change the main messages of the paper or introduce substantial new material (e.g., no new results). Specifically:
> > >
> > > **W1:** We clarified the question about an ambiguous sentence in Figure 2’s caption: each ablation configuration is run repeatedly over the same ablation set of 100 problems.
> > >
> > > **W2:** We addressed the concern about conducting a fair comparison with Hilbert with equalized costs and computational budget. We only restate pass@1 data on PutnamBench reported in their paper together with our original results, e. g.:
> > >
> > > | |Success rate | Token budget per problem |
> > > |--------|--------|-------|
> > > |AxProverBase	|	54.7%	|	5.4 M |
> > > |Hilbert		|	55.9%	|	1880 M |
> > >
> > > **W3:** We clarified the question about the meaning of “cost” in Figure 3(b): it’s the money spent running each of the models on the dataset, normalized with respect to the most expensive experiment.
> > >
> > > **W4:** We addressed the request to expand the quantitative analysis of the failure modes of the system to support claims such as Gemini models hallucinating imports. We provide the data to back up our statements from our original results:
> > >
> > > | Model          | Fraction (%) |
> > > |-----------------|------------------------|
> > > | Gemini Flash       | 56.1         |
> > > | Gemini Pro          | 32.4         |
> > > | Claude Sonnet         | 1.4        |
> > > | Claude Opus        | 0.4          |
> > >
> > > **W5:** We answered the question about the relationship between AxProverBase and the Ax-Prover that we cite: they share the high-level philosophy, but differ substantially in implementation and systematic analysis.
> > >
> > > **W6:** We corrected the minor naming inconsistency between LeanCAT and LeanCat.
> > >
> > > We would be happy to provide additional details to fully address any remaining concerns on these matters.
> > >
> > > # Framework-model performance disentanglement
> > >
> > > We are also happy to comment on the new concern about disentangling the contribution of the agent framework from the underlying frontier model.
> > >
> > > We separate these two factors in our original submission. In Section 4.1 (Figure 2), we study the effect of the agentic framework component (feedback, memory, tools) for a fixed underlying model (Opus 4.5). Thus, the changes in performance should be attributed to the framework, and not to the LLM. Conversely, we apply a fixed framework to several foundation models in Section 4.2, and we observe that they all improve when embedded in the agentic loop (Figure 3(a)), with stronger models benefitting more:
> > >
> > > | Model | Pass@1 | Pass@20 | AxProverBase |
> > > |-------|--------|---------|--------------|
> > > | Gemini 3 Flash | 0.02 | 0.05 | 0.10 |
> > > | Gemini 3 Pro | 0.02 | 0.05 | 0.17 |
> > > | Claude Sonnet 4.5 | 0.01 | 0.04 | 0.29 |
> > > | Claude Opus 4.5 | 0.03 | 0.07 | 0.56 |
> > >
> > > The effect of the underlying model is further analyzed in Figure 3(b), where we compare 12 LLMs in a broad spectrum of model capabilities combining different models and their configurations.
> > >
> > > We agree that we obtain the best results with Claude Opus 4.5 and we do not intend to claim that our framework alone explains the performance independently of the model. With our analysis, we evaluate how much the framework helps at fixed model capacity (Figure 2), and how model improvements interact with the framework (Figure 3), and indeed a key takeaway is that the framework does not remove the dependence on model. The fact that our architecture achieves competitive results with Opus is, in our view, an additional practical result rather than the main focus of the paper. While it raises the significance, the core value of the paper would remain if the performance was lower. In the reviewed version, we will ammend any wording that could be read as an "architecture-only SOTA" claim, and ensure that we frame the Opus 4.5 results as the strongest practical instantiation of the baseline.
> > >
> > > We hope to have addressed your questions about framework-model performance disentanglement. At the moment, it is not clear to us how adding an additional 13th data point in Figure 3 would help address remaining open questions about model-architecture contributions. However, we would be happy to include a Qwen run in the camera-ready version of the paper if further details are provided about these concerns.
> > >
> > > With this submission, we provide a reproducible baseline that could serve as the natural reference for evaluating more complex architectures. As this baseline improves alongside frontier LLMs, it progressively raises the floor for future contributions to the field. This allows us to disentangle the source of improvement: architecture innovation vs access to generally better models, and study how models benefit from such innovations. We therefore hope that the paper is evaluated in this regard.

---

### Official Review · Reviewer_QuQS · 2026-03-09

**Soundness:** 2
**Presentation:** 2
**Significance:** 2
**Originality:** 3
**Overall Recommendation:** 4
**Confidence:** 3

**Summary:**

The paper presents a simple theorem architecture composed of a (1) proposer that, on the basis of the input theorem to be proved, defines a proof strategy, (2) a reviewer that validates the proof being built and (3) a memory which stores the context within which the proof is being built. additionally the proposer is given access to previous tools publicly available. The system is evaluated on various datasets against various state-of-the-art systems, and it consistently shows the best performance

**Compliance With Llm Reviewing Policy:**

Affirmed.

**Key Questions For Authors:**

You should provide more details, making clear where the real added value lies, in particular with reference to how memory and tools work. The impact of these two elements should be clearly evaluated, maybe also with some kind of  extended ablation mechanism

**Limitations:**

The authors describe how to extend the proposed system. They should describe the limitations of the usage of the memory and of the search. How much are these choices arbitrary? What is their impact?

**Strengths And Weaknesses:**

strengths
- The approach is simple and intuitive
- The proposed approach performs very well over the state-of-the-art systems
- The proposed architecture seems to capture a rather general approach for organising reasoning.

weaknesses
- The real reason why this approach works is somehow not clear. Clearly, the memory mechanism heuristics highly impact the final result. The library and the web search potentially provide access to a lot of reasoning strategies. What are the heuristics which choose what to remember inside the memory mechanism? What are the criteria used to decide what the search functionality is given access to? What is the key factor for the success of the proposed approach?

---

> ### Author Rebuttal · Authors · 2026-03-31
>
> Thank you for carefully reading our paper and for asking us to clarify where the added value lies, especially regarding memory and tools.
>
> At a high level, the added value of this paper lies in isolating the core components of modern provers and systematically measuring their impact. As you noted, it is not always clear why complex provers work. Our controlled bottom-up ablation answers this directly: the key factor for success is iterative refinement driven by Lean compiler feedback, while memory is the second important factor and tools add a smaller gain.
>
> As shown in Figures 2 and 3(a), repeated independent sampling (pass@k) is highly inefficient and providing information on the previously failed attempts drastically improves performance. Just providing feedback from the last attempt already gives the largest gain (see Fig.2). To enhance this even further, we explore a memory mechanism to manage the agent’s context.
>
> A naive memory design would keep the full history of all attempts, quickly saturating the context of the underlying LLM. Therefore, we explore the following alternatives to keep context under control:
> 1. Feedback from last attempt
> 2. Truncated history with the most recent attempts, including the agent’s reasoning, resulting code, and reviewer feedback.
> 3. Self-managed context, where the agent itself reflects on the outcome of every proof attempt and keeps track of the information that may help in future iterations, similarly to having a scratchpad where the agent can take notes on past attempts. We do not restrict what can be stored, and rely on prompting for giving guidelines on what to include. The agent decides what to keep and how to structure it.
>
> A priori, there is no evident advantage of one memory strategy with respect to the other, and we resort to systematic experimentation to select based on empirical evaluation of the performance that they provide. Out of these, we observe that self-managed context yields the strongest results. This method can compress the most relevant information and preserve it for many iterations, at the expense of an additional LLM call.
>
> Regarding tools, the choice was motivated by current state-of-the-art. Many provers in literature are equipped with Mathlib search and web search, where especially Mathlib search is considered one of the critical features, given that this library plays a central role in most mathematical applications. Hence, we consider these tools in our study and we evaluate their contribution to the overall performance.
>
> One of the most important empirical observations is that having these tools is useful, but not as nearly as significant as introducing a memory mechanism with a compiler feedback. The jump in Fig. 2 from single-shot to feedback is large, while the one from self-reflection to full system is relatively small. Our hypothesis is that modern flagship models, like Claude Opus 4.5, are good at solving complex mathematics problems (hence web search is not crucial to get an informal solution) and have good intrinsic knowledge of Mathlib (hence Mathlib search is not so impactful), but are mainly limited by lack of interaction with Lean compiler. Investigating tool use in detail is not trivial and deserves a separate study, which would be out of scope for this paper, but is enabled to the community by our open-source architecture.
>
> Our study contributes to the field in multiple ways. First, it shows that a simple agentic approach can compete against much more complex architectures, calling for careful choice and optimization of future agentic systems for theorem proving. Secondly, we set an open-source reference baseline that others can build upon. Finally, we provide a systematic study of the aforementioned key success factors, i.e. the memory mechanism and the iterative refinement.
> The formal theorem proving community also benefits from this work. We found that iterative MCP calling, popular among Lean4 developers, is suboptimal for complex proofs due to poor context management (Fig. 2). Secondly, as we show in the reply to Reviewer 1, our open-source agent is much more cost effective than some other alternatives and can be readily adopted in real-world projects.
>
> Our reviewed version of the manuscript includes:
> - A direct comparison of cost-performance with state of the art methods where we show that we achieve the same pass@1 performance as Hilbert on PutnamBench using about 200x less tokens.
> - A cost-performance analysis of our approach for all datasets including the pareto frontier of success rate vs tokens.
> - A quantitative analysis of the failure modes. E. g., 56% and 32% of the Gemini Flash and Pro errors contain hallucinated imports.
> - Various improvements in the formatting and discussion sections.
>
> Overall, we believe the paper has significantly improved in the quality of its exposition and in the significance of its results (see the second paragraph in the Reviewer 1 rebuttal for more details).

---

> > ### Author Rebuttal · Reviewer_QuQS · 2026-04-04
> >
> > there were no major concernes to begin with

---

### Official Review · Reviewer_YzuR · 2026-03-11

**Soundness:** 2
**Presentation:** 2
**Significance:** 2
**Originality:** 2
**Overall Recommendation:** 4
**Confidence:** 3

**Summary:**

This paper presents AxProverBase, a minimal agentic baseline for automated theorem proving in Lean 4. The system is composed of three modular components: a proposer agent (a general-purpose LLM using a ReAct-style loop), a review system (compiler + LLM-based reviewer to catch cheating tactics like `native_decide`), and a memory module (with variants including no memory, history of past k attempts, and self-managed context via self-reflection). The proposer can optionally call tools such as LeanSearch for Mathlib premise retrieval and Tavily for web search. Through a systematic bottom-up ablation study on a 100-problem subset of PutnamBench, the authors show that iterative proof refinement is the single largest contributor to performance, followed by the self-managed memory mechanism, with tool access providing a smaller marginal gain. They further compare multiple foundation models (Claude Opus/Sonnet 4.5, Gemini 3 Flash/Pro) and analyze the effect of thinking budgets. The full system, using Claude Opus 4.5 with 32k thinking tokens and 50 iterations, is evaluated on PutnamBench, FATE, and LeanCat, achieving competitive results (e.g., 54.7% pass@1 on PutnamBench, 98.0% pass@1 on FATE-M) compared to far more complex state-of-the-art systems that rely on specialized training or reinforcement learning.

**Compliance With Llm Reviewing Policy:**

Affirmed.

**Final Justification:**

All of my concerns have been addressed, so I have increased my score.

**Key Questions For Authors:**

1. Can you provide a concrete cost and execution time comparison with at least one or two external systems (e.g., Goedel-Prover V2) on the same benchmark, to substantiate the claim of being significantly more efficient?

2. How sensitive are the ablation conclusions to the choice of the 100-problem subset? Have you verified that the key finding (iterative refinement > memory > tools) holds on a different random subset or on the full PutnamBench?

3. Given that the system's strong performance may largely stem from Claude Opus 4.5's capabilities, have you considered evaluating with a weaker but more accessible open-source model (e.g., a Qwen variant) to better isolate the architectural contribution from the model contribution?

**Limitations:**

Yes

**Strengths And Weaknesses:**

## Strengths

1. Well-motivated and practically useful contribution. The automated theorem proving field is increasingly crowded with complex systems where it is difficult to disentangle the sources of improvement. A clean, modular baseline that isolates the effect of each component (feedback, memory, tools, model choice) fills a genuine gap and provides a valuable reference point for future research.

2. Clear ablation design. The bottom-up experimental methodology, starting from single-shot and progressively adding iterative refinement, memory, and tools, makes the marginal contribution of each component clearly visible. The finding that iterative refinement alone (pass@1) can outperform single-shot approaches at high sample counts (pass@k) is a useful quantitative confirmation that is well-supported by the data in Figure 2.

3. Broad evaluation scope. The paper evaluates on three qualitatively different benchmarks (competition math via PutnamBench, research-level abstract algebra via FATE, and category theory via LeanCat), demonstrating that the simple agentic approach generalizes across mathematical domains without domain-specific tuning.

4. Insightful model comparison. The cross-model analysis (Section 4.2) reveals that Claude models prove roughly 3x more theorems than Gemini counterparts, along with the diagnosis that Gemini models tend to hallucinate outdated Mathlib imports, providing actionable insights for practitioners choosing foundation models for theorem proving.

## Weaknesses

1. Limited technical novelty. The individual components, iterative proof refinement with compiler feedback, memory/self-reflection, library search, and LLM-based review, are all well-established in prior work (Goedel-Prover V2, Seed-Prover, Hilbert, etc.). The paper's contribution is primarily in assembling and ablating these components, which, while useful, is incremental. The architecture section (Section 3) is relatively thin and does not introduce any new techniques.

2. Unfair and hard-to-interpret comparison in Table 1. The main comparison table juxtaposes systems with vastly different computational budgets: pass@1 (AxProverBase) vs. pass@1024 (DeepSeek V2) vs. pass@1840 (Hilbert). Different backbone models, training regimes (RL, fine-tuning vs. zero-shot prompting), and cost profiles make direct numerical comparison misleading. The paper would benefit from a normalized comparison, e.g., performance at matched compute cost, or a Pareto frontier of cost vs. accuracy across systems. Additionally, the table overflows the column width.

3. Lack of efficiency and cost comparison with external systems. For a paper that emphasizes accessibility and cost-effectiveness, the cost analysis is surprisingly shallow. The claim of being "an order of magnitude" faster than Hilbert (Section 4.3) is stated without supporting data. A table comparing execution time, token consumption, and dollar cost per solved problem across systems would substantially strengthen the paper's practical message.

4. Ablations on a small subset and incomplete presentation. All ablation studies are conducted on only 100 randomly sampled PutnamBench problems, and the confidence intervals in Figure 2 are relatively wide, which limits the reliability of fine-grained conclusions (e.g., the relative benefit of history vs. self-reflection). More importantly, the paper lacks a Conclusion section, the main body is only 7 pages, and the Impact Statement is perfunctory, collectively giving the impression of an incomplete manuscript.

---

> ### Author Rebuttal · Authors · 2026-03-31
>
> Thank you for the careful review and for recognizing the value of a clean, modular baseline for automated theorem proving. We appreciate your feedback, which has helped us significantly strengthen the paper. Let us address it point by point.
>
> **On Limited Technical Novelty (Weakness 1):**
> We agree the individual components are not novel ideas on their own. However, as you noted, current state-of-the-are does not disentangle their impact, which is our core contribution. Our controlled, bottom-up study isolates what drives performance: iterative refinement is the dominant factor, memory design is second, and search tools provide a smaller but positive improvement. Our study contributes to the broader field through providing:
> - A reality check for architecture design: We prove a simple agent rivals complex architectures, establishing a rigorous baseline.
> - Practical insights for Lean developers: We show that iterative agents with tools with unrestricted context, currently popular among Lean 4 developers, may actually be suboptimal for complex proofs due to poor context management (Fig. 2).
> - A cost-effective tool: Our agent is efficient and cost-effective (see below). We release it open-source, so it can be readily adopted for real-world formalization projects.
>
> **On Fair Comparison, Efficiency, and Cost (Weaknesses 2 & 3, Question 1):**
> At submission, we lacked normalized metrics across the literature for fair comparison. Fortunately, recent revisions of ICLR-accepted publications now provide these details. Compared with Hilbert over PutnamBench, we match its pass@1 success rate with ~200x less tokens. Comparing DeepSeek R1, DeepSeek Prover V2, Qwen3, Goedel Prover V2 and Kimina Prover in the FATE (M and H) datasets, we substantially outperformed them at matched token budgets. We believe these new results significantly strengthen our main points with direct evidence, such as AxProverBase and Hilbert respectively achieving a pass@1 success rate in PutnamBench of 54.7% and 55.9% with 5.4 and 1880.4 max million tokens per problem. See the new supporting figures [here](https://anonymous.4open.science/r/icml-axproverbase/comparison.md).
> Distilling the main contributing elements to performance, allowed us to achieve competitive results at a fraction of the cost, showing that the field would significantly benefit from strong baselines that capitalize on the progress of frontier models and against which complex architectures should be evaluated.
>
> Furthermore, we have included an additional analysis about performance vs cost of our method (pareto frontier) for all the datasets. We also separate input and output tokens given that they imply significantly different costs.
>
> **On Ablation Subset Sensitivity (Weakness 4, Question 2):**
> The goal of our ablation study is to show that every element in the system contributes to its performance. As we mention in the paper, we randomly sub-sample 100 out of 660 problems to limit the cost (allowing us to conduct more experiments), and to prevent overfitting to a particular dataset, given that we select our architecture hyper parameters based on the result. We verified that the qualitative ordering in the ablation (feedback > memory > tools) is stable across multiple random subsamples through a statistical analysis. Additionally, the “full system” line calculated on the ablation dataset is consistent with the analogous run on the full dataset of 660 problems. See data [here](https://anonymous.4open.science/r/icml-axproverbase/ablation.md).
>
> **On Evaluating Open-Source Models like Qwen (Question 3):**
> While supporting open-source models broadens adoption, Section 4.2 already compares LLMs with varying capabilities. Single-shot, these models hover around ~5% success. Embedded in our system, performance ranges widely (10-55%), with stronger models benefiting the most. Achieving competitive performance with Opus 4.5 via simple scaffolding is a remarkable result pass@k could not predict.
>
> Regarding Qwen, we note that strong performance on structured output and function calling is critical for this task. However, current implementations of self-hosted Qwen3 struggle with structured output parsing on vLLM. In addition, when combining thinking mode with tool calling, the tool calls are not detected, and the hosted Alibaba version has strong rate limits that prevent systematic experiments. We agree that supporting open source models would broaden adoption, but reliable evaluation currently has nontrivial engineering overhead, which we leave for future work. Find a list of known issues [here](https://anonymous.4open.science/r/icml-axproverbase/qwen.md).
>
> **On Presentation and Formatting (Weakness 4):**
> Thank you for catching the formatting issues. For the camera-ready revision, we have fixed the Table 1 overflow, added a dedicated Conclusion section summarizing our findings, and revised the Impact Statement.

---

> > ### Author Rebuttal · Reviewer_YzuR · 2026-04-03
> >
> > Thank you for your response. I have no further questions.

---

### Decision · Program_Chairs · 2026-04-30

**Decision:**

Accept (regular)

**Comment:**

Reviewers agree this is a solid empirical study with a clean, modular baseline and clear ablations isolating key components (iteration, memory, tools). While novelty is limited, the work provides useful insights and a strong reference point for the community. Most concerns (e.g., evaluation clarity, cost analysis) were addressed in the rebuttal. Overall, a practical and valuable contribution.